# Analysis and Design of Thompson Sampling for Stochastic Partial Monitoring

**Taira Tsuchiya**
The University of Tokyo
RIKEN AIP
tsuchiya@ms.k.u-tokyo.ac.jp

**Junya Honda**
The University of Tokyo
RIKEN AIP
honda@edu.k.u-tokyo.ac.jp

**Masashi Sugiyama**
RIKEN AIP
The University of Tokyo
sugi@k.u-tokyo.ac.jp

## Abstract

We investigate finite stochastic partial monitoring, which is a general model for sequential learning with limited feedback. While Thompson sampling is one of the most promising algorithms on a variety of online decision-making problems, its properties for stochastic partial monitoring have not been theoretically investigated, and the existing algorithm relies on a heuristic approximation of the posterior distribution. To mitigate these problems, we present a novel Thompson-sampling-based algorithm, which enables us to exactly sample the target parameter from the posterior distribution. Besides, we prove that the new algorithm achieves the logarithmic *problem-dependent expected pseudo-regret* $\mathrm{O}(\log T)$ for a linearized variant of the problem with local observability. This result is the first regret bound of Thompson sampling for partial monitoring, which also becomes the first logarithmic regret bound of Thompson sampling for linear bandits.

## 1 Introduction

Partial monitoring (PM) is a general sequential decision-making problem with limited feedback (Rustichini, 1999; Piccolboni and Schindelhauer, 2001). PM is attracting broad interest because it includes a wide range of problems such as the multi-armed bandit problem (Lai and Robbins, 1985), a linear optimization problem with full or bandit feedback (Zinkevich, 2003; Dani et al., 2008), dynamic pricing (Kleinberg and Leighton, 2003), and label efficient prediction (Cesa-Bianchi et al., 2005).

A PM game can be seen as a sequential game that is played by two players: a learner and an opponent. At every round, the learner chooses an action, while the opponent chooses an outcome. Then, the learner suffers an unobserved loss and receives a feedback symbol, both of which are determined from the selected action and outcome. The main characteristic of this game is that the learner cannot directly observe the outcome and loss. The goal of the learner is to minimize his/her cumulative loss over all rounds. The performance of the learner is evaluated by the regret, which is defined as the difference between the cumulative losses of the learner and the optimal action (*i.e.,* the action whose expected loss is the smallest).

There are mainly two types of PM games, which are the *stochastic* and *adversarial* settings (Piccolboni and Schindelhauer, 2001; Bartók et al., 2011). In the stochastic setting, the outcome at each round is determined from the *opponent's strategy*, which is a probability vector over the opponent's possible choices. On the other hand, in the adversarial setting, the outcomes are arbitrarily decided by the

opponent. We refer to the PM game with finite actions and finite outcomes as a *finite* PM game. In this paper, we focus on the stochastic finite game.

One of the first algorithms for PM was considered by Piccolboni and Schindelhauer (2001). They proposed the FeedExp3 algorithm, the key idea of which is to use an unbiased estimator of the losses. They showed that the FeedExp3 algorithm attains $\tilde{O}(T^{3/4})$ minimax regret for a certain class of PM games, and showed that any algorithm suffers linear minimax regret $\Omega(T)$ for the other class. Here $T$ is the time horizon and the notation $\tilde{O}(\cdot)$ hides polylogarithmic factors. The upper bound $\tilde{O}(T^{3/4})$ is later improved by Cesa-Bianchi et al. (2006) to $O(T^{2/3})$, and they also provided a game with a matching lower bound.

In the seminal paper by Bartók et al. (2011), they classified PM games into four classes based on their minimax regrets. To be more specific, they classified games into trivial, easy, hard, and hopeless games, where their minimax regrets are $0$, $\tilde{\Theta}(\sqrt{T})$, $\Theta(T^{2/3})$, and $\Theta(T)$, respectively. Note that the easy game is also called a *locally observable* game. After their work, several algorithms have been proposed for the finite PM problem (Bartók et al., 2012; Vanchinathan et al., 2014; Komiyama et al., 2015). For the problem-dependent regret analysis, Komiyama et al. (2015) proposed an algorithm that achieves $O(\log T)$ regret with the optimal constant factor. However, it requires to solve a time-consuming optimization problem with infinitely many constraints at each round. In addition, this algorithm relies on the forced exploration to achieve the optimality, which makes the empirical performance near-optimal only after prohibitively many rounds, say, $10^5$ or $10^6$.

Thompson sampling (TS, Thompson, 1933) is one of the most promising algorithms on a variety of online decision-making problems such as the multi-armed bandit (Lai and Robbins, 1985) and the linear bandit (Agrawal and Goyal, 2013b), and the effectiveness of TS has been investigated both empirically (Chapelle and Li, 2011) and theoretically (Kaufmann et al., 2012; Agrawal and Goyal, 2013a; Honda and Takemura, 2014). In the literature on PM, Vanchinathan et al. (2014) proposed a TS-based algorithm called BPM-TS (Bayes-update for PM based on TS) for stochastic PM, which empirically achieved state-of-the-art performance. Their algorithm uses Gaussian approximation to handle the complicated posterior distribution of the opponent's strategy. However, this approximation is somewhat heuristic and can degrade the empirical performance due to the discrepancy from the exact posterior distribution. Furthermore, no theoretical guarantee is provided for BPM-TS.

Our goals are to establish a new TS-based algorithm for stochastic PM, which allows us to sample the opponent's strategy parameter from the exact posterior distribution, and investigate whether the TS-based algorithm can achive sub-linear regret in stochastic PM. Using the accept-reject sampling, we propose a new TS-based algorithm for PM (TSPM), which is equipped with a numerical scheme to obtain a posterior sample from the complicated posterior distribution. We derive a logarithmic regret upper bound $O(\log T)$ for the proposed algorithm on the locally observable game under a linearized variant of the problem. This is the first regret bound for TS on the locally observable game. Moreover, our setting includes the linear bandit problem and our result is also the first logarithmic expected regret bound of TS for the linear bandit, whereas a high-probability bound was provided, for example, in Agrawal and Goyal (2013b). Finally, we compare the performance of TSPM with existing algorithms in numerical experiments, and show that TSPM outperforms existing algorithms.

## 2 Preliminaries

This paper studies finite stochastic PM games (Bartók et al., 2011). A PM game with $N$ actions and $M$ outcomes is defined by a pair of a loss matrix $\mathbf{L} = (\ell_{i,j}) \in \mathbb{R}^{N \times M}$ and feedback matrix $\mathbf{H} = (h_{i,j}) \in [A]^{N \times M}$, where $A$ is the number of feedback symbols and $[A] = \{1, 2, \dots, A\}$.

A PM game can be seen as a sequential game that is played by two players: the learner and the opponent. At each round $t = 1, 2, \dots, T$, the learner selects action $i(t) \in [N]$, and at the same time the opponent selects an outcome based on the opponent's strategy $p^* \in \mathcal{P}_M$, where $\mathcal{P}_n = \{p \in \mathbb{R}^n : p_k \geq 0, \sum_{k=1}^n p_k = 1\}$ is the $(n-1)$-dimensional probability simplex. The outcome $j(t)$ of each round is an independent and identically distributed sample from $p^*$, and then, the learner suffers loss $\ell_{i(t),j(t)}$ at time $t$. The learner cannot directly observe the value of this loss, but instead observes the *feedback symbol* $y(t) = h_{i(t),j(t)} \in [A]$. The setting explained above has been widely studied in the literature of stochastic PM (Bartók et al., 2011; Komiyama et al., 2015), and we call this the *discrete* setting. In Section 4, we also introduce a *linear* setting for theoretical analysis, which is a slightly different setting from the discrete one.

The learner aims to minimize the cumulative loss over $T$ rounds. The expected loss of action $i$ is given by $L_i^\top p^*$, where $L_i$ is the $i$-th column of $\mathbf{L}^\top$. We say action $i$ is *optimal* under strategy $p^*$ if $(L_i - L_j)^\top p^* \leq 0$ for any $j \neq i$. We assume that the optimal action is unique, and without loss of generality that the optimal action is action 1. Let $\Delta_i = (L_i - L_1)^\top p^* \geq 0$ for $i \in [N]$ and $N_i(t)$ be the number of times action $i$ is selected before the $t$-th round. When the time step is clear from the context, we use $n_i$ instead of $N_i(t)$. We adopt the pseudo-regret to measure the performance: $\mathrm{Reg}(T) = \sum_{t=1}^T \Delta_{i(t)} = \sum_{i \in [N]} \Delta_i N_i(T+1)$. This is the relative performance of the algorithm against the *oracle*, which knows the optimal action 1 before the game starts.

We introduce the following definitions to clarify the class of PM games, for which we develop an algorithm and derive a regret upper bound. The following cell decomposition is the concept to divide the simplex $\mathcal{P}_M$ based on the loss matrix to identify the optimal action, which depends on the opponent's strategy $p^*$.

**Definition 1** (Cell decomposition and Pareto-optimality (Bartók et al., 2011))**.** For every action $i \in [N]$, *cell* $\mathcal{C}_i := \{p \in \mathcal{P}_M : (L_i - L_j)^\top p \leq 0, \forall j \neq i\}$ is the set of opponent's strategies for which action $i$ is optimal. Action $i$ is *Pareto-optimal* if there exists an opponent's strategy $p^*$ under which action $i$ is optimal.

Each cell is a convex closed polytope. Next, we define *neighbors* between two Pareto-optimal actions, which intuitively means that the two actions "touch" each other in their surfaces.

**Definition 2** (Neighbors and neighborhood action (Bartók et al., 2011))**.** Two Pareto-optimal actions $i$ and $j$ are *neighbors* if $\mathcal{C}_i \cap \mathcal{C}_j$ is an $(M-2)$-dimensional polytope. For two neighboring actions $i, j \in [N]$, the *neighborhood action set* is defined as $N_{i,j}^+ = \{k \in [N] : \mathcal{C}_i \cap \mathcal{C}_j \subseteq \mathcal{C}_k\}$.

Note that the neighborhood action set $N_{i,j}^+$ includes actions $i$ and $j$ from its definition. Next, we define the *signal matrix*, which encodes the information of the feedback matrix $\mathbf{H}$ so that we can utilize the feedback information.

**Definition 3** (Signal matrix (Komiyama et al., 2015))**.** The signal matrix $S_i \in \{0,1\}^{A \times M}$ of action $i$ is defined as $(S_i)_{y,j} = \mathbb{1}[h_{i,j} = y]$, where $\mathbb{1}[X] = 1$ if the event $X$ is true and $0$ otherwise.

Note that if we define the signal matrix as above, $S_i p^* \in \mathbb{R}^A$ is a probability vector over feedback symbols of action $i$. The following *local observability* condition separates easy and hard games, this condition intuitively means that the information obtained by taking actions in the neighborhood action set $N_{i,j}^+$ is sufficient to distinguish the loss difference between actions $i$ and $j$.

**Definition 4** (Local observability (Bartók et al., 2011))**.** A partial monitoring game is said to be *locally observable* if for all pairs $i, j$ of neighboring actions, $L_i - L_j \in \oplus_{k \in N_{i,j}^+} \mathrm{Im}\, S_k^\top$, where $\mathrm{Im}\, V$ is the image of the linear map $V$, and $V \oplus W$ is the direct sum between the vector spaces $V$ and $W$.

We also consider the concept of the *strong local observability* condition, which implies the above local observability condition.

**Definition 5** (Strong local observability)**.** A partial monitoring game is said to be *strongly locally observable* if for all pairs $i, j \in [N]$, $L_i - L_j \in \mathrm{Im}\, S_i^\top \oplus \mathrm{Im}\, S_j^\top$.

This condition was assumed in the theoretical analysis in Vanchinathan et al. (2014), and we also assume this condition in theoretical analysis in Section 4. Note that the strong local observability means that, for any $j \neq k$, there exists $z_{j,k} \neq 0 \in \mathbb{R}^{2A}$ such that $L_j - L_k = (S_j^\top, S_k^\top) z_{j,k}$.

**Notation.** Let $\|\cdot\|$ and $\|\cdot\|_p$ be the Euclidian norm and $p$-norm, and let $\|x\|_A = \sqrt{x^\top A x}$ be the norm induced by the positive semidefinite matrix $A \succeq 0$. Let $\mathcal{D}_{\mathrm{KL}}(p\|q) = \sum_{a=1}^A p_a \log(p_a/q_a)$ be the Kullback-Leibler divergence of $p$ from $q$. The vector $e_y \in \mathbb{R}^M$ is the $y$-th orthonormal basis of $\mathbb{R}^M$, and $\mathbf{1}_n = [1, \ldots, 1]^\top$ is the $n$-dimensional all-one vector. Let $q_i^{(t)}$ be the empirical feedback distribution of action $i$ at time $t$, *i.e.*, $q_i^{(t)} = [n_{i1}/n_i, \ldots, n_{iA}/n_i]^\top \in \mathcal{P}_A$, where $n_{iy} = \sum_{s=1}^t \mathbb{1}[i(s) = i, y(s) = y]$ and $n_i = \sum_{y=1}^A n_{iy}$. The notation is summarized in Appendix A.

**Methods for Sampling from Posterior Distribution.** We briefly review the methods to draw a sample from the posterior distribution. While TS is one of the most promising algorithms, the posterior distribution can be in a quite complicated form, which makes obtaining a sample from it

**Algorithm 1:** TSPM Algorithm
_____________________________________________________________________
**Input:** prior parameter $\lambda > 0$
1  Set $B_0 \leftarrow \lambda I_M, b_0 \leftarrow 0$.
2  Take each action for $n \geq 1$ times.
3  **for** $t = 1, 2, \ldots, T$ **do**
4  $\quad$ Sample $\tilde{p}_t \sim \pi(p \mid \{i(s), y(s)\}_{s=1}^t)$ based on the accept-reject sampling (Algorithm 2).
5  $\quad$ Take action $i(t) = \arg\max_{i \in [N]} L_i^\top \tilde{p}_t$ and observe feedback $y(t)$.
6  $\quad$ Update $B_t \leftarrow B_{t-1} + S_{i(t)}^\top S_{i(t)}, \; b_t \leftarrow b_{t-1} + S_{i(t)}^\top e_{y(t)}$.

**Algorithm 2:** Accept-Reject Sampling
_____________________________________________
**Input:** constant $R \in [0, 1]$
1  **while** true **do**
2  $\quad$ Sample $\tilde{p}_t \sim g_t(p)$ (Algorithm 3).
3  $\quad$ Sample $\tilde{u} \sim \mathcal{U}([0, 1])$.
4  $\quad$ **if** $R\tilde{u} < F_t(\tilde{p}_t)/G_t(\tilde{p}_t)$ **then**
5  $\quad\quad$ **return** $\tilde{p}_t$.

**Algorithm 3:** Sampling from $g_t(p)$
_____________________________________________
1  Compute $\tilde{B}_t, \tilde{b}_t$ from $B_t, b_t$.
2  **repeat**
3  $\quad$ Sample $p^{(\alpha)} \sim \mathcal{N}(\tilde{B}_t^{-1}\tilde{b}_t, \tilde{B}_t^{-1})$.
4  **until** $p^{(\alpha)} \in \mathcal{P}_{M-1}$ ;
5  **return** $\tilde{p} = [{p^{(\alpha)}}^\top, \, 1 - \sum_{i=1}^{M-1}(p^{(\alpha)})_i]^\top$.

computationally hard. To overcome this issue, a variety of approximate posterior sampling methods have been considered, such as Gibbs sampling, Langevin Monte Carlo, Laplace approximation, and the bootstrap (Russo et al., 2018, Section 5). Recent work (Lu and Van Roy, 2017) proposed a flexible approximation method, which can even efficiently be applied to quite complex models such as neural networks. However, more recent work revealed that algorithms based on such an approximation procedure *can* suffer a linear regret (Phan et al., 2019), even if the approximation error in terms of the $\alpha$-divergence is small enough.

Although BPM-TS is one of the best methods for stochastic PM, it approximates the posterior by a Gaussian distribution in a heuristic way, which can degrade the empirical performance due to the distributional discrepancy from the exact posterior distribution. Furthermore, no theoretical guarantee is provided for BPM-TS. In this paper, we mitigate these problems by providing a new algorithm for stochastic PM, which allows us to exactly draw samples from the posterior distribution. We also give theoretical analysis for the proposed algorithm.

## 3  Thompson-sampling-based Algorithm for Partial Monitoring

In this section, we present a new algorithm for stochastic PM games, where we name the algorithm TSPM (TS-based algorithm for PM). The algorithm is given in Algorithm 1, and we will explain the subroutines in the following.

### 3.1  Accept-Reject Sampling

We adopt the accept-reject sampling (Casella et al., 2004) to *exactly* draw samples from the posterior distribution. The accept-reject sampling is a technique to draw samples from a specific distribution $f$, and a key feature is to use a *proposal distribution* $g$, from which we can easily draw a sample and whose ratio to $f$, that is $f/g$, is bounded by a constant value $R$. To obtain samples from $f$, (i) we generate samples $X \sim g$; (ii) accept $X$ with probability $f(X)/Rg(X)$. Note that $f$ and $g$ do not have to be normalized when the acceptance probability is calculated.

Let $\pi(p)$ be a prior distribution for $p$. Then an unnormalized density of the posterior distribution for $p$ can be expressed as

$$F_t(p) = \pi(p) \prod_{i=1}^N \exp\left\{ -n_i \mathcal{D}_{\mathrm{KL}}\left( q_i^{(t)} \middle\| S_i p \right) \right\}, \tag{1}$$

the detailed derivation of which is given in Appendix B. We use the proposal distribution with unnormalized density

$$G_t(p) = \pi(p) \prod_{i=1}^{N} \exp\left\{-\frac{1}{2} n_i \|q_i^{(t)} - S_i p\|^2\right\}. \tag{2}$$

Based on these distributions, we use Algorithm 2 for exact sampling from the posterior distribution, where $\mathcal{U}([0,1])$ is the uniform distribution over $[0,1]$ and $g_t(p)$ is the distribution corresponding to the unnormalized density $G_t(p)$ in (2). The following proposition shows that setting $R = 1$ realizes the exact sampling.

**Proposition 1.** *Let $f_t(p)$ be the distribution corresponding to the unnormalized density $F_t(p)$ in (1). Then, the output of Algorithm 2 with $R = 1$ follows $f_t(p)$.*

This proposition can easily be proved by Pinsker's inequality, which is detailed in Appendix B.

In practice, $R \in [0,1]$ is a parameter to balance the amount of over-exploration and the computational efficiency. As $R$ decreases from 1, the algorithm tends to accept a point $p$ far from the mode. The case $R = 0$ corresponds the TSPM algorithm where the proposal distribution is used without the accept-reject sampling, which we call *TSPM-Gaussian*. As we will see in Section 4, TSPM-Gaussian corresponds to exact sampling of the posterior distribution when the feedback follows a Gaussian distribution rather than a multinomial distribution.

TSPM-Gaussian can be related to BPM-TS (Vanchinathan et al., 2014) in the sense that both of them use samples from Gaussian distributions. Nevertheless, they use different Gaussians and TSPM-Gaussian performs much better than BPM-TS as we will see in the experiments. Details on the relation between TSPM-Gaussian and BPM-TS are described in Appendix D.

In general, we can realize efficient sampling with a small number of rejections if the proposal distribution and the target distribution are close to each other. On the other hand, in our problem, the densities in (1) and (2) for each fixed point $p$ exponentially decay with the number of samples $n_i$ if the empirical feedback distribution $q_i^{(t)}$ converges. This means that $F_t(p)$ and $G_t(p)$ have an exponentially large relative gap in most rounds. Nevertheless, the number of rejections does not increase with $t$ as we will see in the experiments, which suggests that the proposal distribution approximates the target distribution well with high probability.

### 3.2  Sampling from Proposal Distribution

When we consider Gaussian density $\mathcal{N}(0, \lambda I_M)$ truncated over $\mathcal{P}_M$ as a prior, the proposal distribution also has the Gaussian density $\mathcal{N}(B_t^{-1} b_t, B_t^{-1})$ over $\mathcal{P}_M$, where

$$B_t = \lambda I_M + \sum_{i=1}^{N} n_i S_i^\top S_i = B_{t-1} + S_{i(t)}^\top S_{i(t)}, \quad b_t = \sum_{i=1}^{N} n_i S_i^\top q_i^{(t)} = b_{t-1} + S_{i(t)}^\top e_{y(t)}. \tag{3}$$

Here note that the probability simplex $\mathcal{P}_M$ is in an $(M-1)$-dimensional space and a sample from $\mathcal{N}(0, \lambda I_M)$ is not contained in $\mathcal{P}_M$ with probability one. In the literature, *e.g.*, Altmann et al. (2014), sampling methods for Gaussian distributions truncated on a simplex have been discussed. We use one of these procedures summarized in Algorithm 3, where we first sample $M-1$ elements of $p$ from another Gaussian distribution and determine the remaining element by the constraint $\sum_{i=1}^{M} p_i = 1$.

**Proposition 2.** *Sampling from $g_t(p)$ is equivalent to Algorithm 3 with*

$$\tilde{B}_t = C_t - 2D_t + f_t \mathbf{1}_{M-1} \mathbf{1}_{M-1}^\top, \quad \tilde{b}_t = f_t \mathbf{1}_{M-1} - d_t + b_t^{(\alpha)} - b^{(M)} \mathbf{1}_{M-1},$$

*where $B_t = \begin{bmatrix} C_t & d_t \\ d_t^\top & f_t \end{bmatrix}$ for $C_t \in \mathbb{R}^{M-1 \times M-1}$, $d_t \in \mathbb{R}^{M-1}$, $f_t \in \mathbb{R}$, $b_t = [b_t^{(\alpha)\top}, b_t^{(M)}]^\top \in \mathbb{R}^{M-1} \times \mathbb{R}$, and $D_t = \frac{1}{2}(d_t \mathbf{1}_{M-1}^\top + \mathbf{1}_{M-1} d_t^\top)$.*

We give the proof of this proposition for self-containedness in Appendix C.

## 4  Theoretical Analysis

This section considers a regret upper bound of the TSPM algorithm.

In the theoretical analysis, we consider a *linear* setting of PM. In the linear PM, the learner suffers the expected loss $L_{i(t)}^\top p^*$ as in the discrete setting, and receives feedback vector $y(t) = S_i p^* + \epsilon_t$ for $\epsilon_t \sim \mathcal{N}(0, I_M)$ whereas the one-hot representation of $y(t)$ is distributed by the probability vector $S_i p^*$ in the discrete setting. Therefore, if $\epsilon_t$ can be regarded as a sub-Gaussian random variable as in Kirschner et al. (2020) then the linear PM includes the discrete PM, though our theoretical analysis requires $\epsilon_t$ to be Gaussian. The relation between discrete and linear settings can also be seen from the observation that bandit problems with Bernoulli and Gaussian rewards can be expressed as discrete and linear PM, respectively. The linear PM also includes the linear bandit problem, where the feedback vector is expressed as $L_i^\top p^* + \epsilon_t$.

In the linear PM, $G_t(p)$ in (2) becomes the exact posterior distribution rather than a proposal distribution. The definition of the cell decomposition for this setting is largely the same as that of discrete setting and detailed in Appendix F. Therefore, TS with exact posterior sampling in the linear PM corresponds to TSPM-Gaussian. In the linear PM, the unknown parameter $p^*$ is in $\mathbb{R}^M$ rather than in $\mathcal{P}_M$, and therefore we consider the prior $\pi(p) = \mathcal{N}(0, \lambda I_M)$ over $\mathbb{R}^M$, where the posterior distribution becomes $\mathcal{N}(B_t^{-1} b_t, B_t^{-1})$.

There are a few works that analyze TS for the PM because of its difficulty. For example in Vanchinathan et al. (2014), an analysis of the TS-based algorithm (BPM-TS) is not given despite the fact that its performance is better than the algorithm based on a confidence ellipsoid (BPM-LEAST). Zimmert and Lattimore (2019) considered the theoretical aspect of a variant of TS for the linear PM in view of the Bayes regret, but this algorithm is based on the knowledge on the time horizon and different from the family of TS used in practice. More specifically, their algorithm considers the posterior distribution for *regret* (not pseudo-regret), and an action is chosen according to the posterior probability that each arm minimizes the *cumulative* regret. Thus, the time horizon also needs to be known.

**Types of Regret Bounds.** We focus on the *(a) problem-dependent (b) expected pseudo-regret.* (a) In the literature, a *minimax* (or *problem-independent*) regret bound has mainly been considered, for example, to classify difficulties of the PM problem (Bartók et al., 2010; Bartók et al., 2011). On the other hand, a *problem-dependent* regret bound often reflects the empirical performance more clearly than the minimax regret (Bartók et al., 2012; Vanchinathan et al., 2014; Komiyama et al., 2015). For this reason, we consider this problem-dependent regret bound. (b) In complicated settings of bandit problems, a *high-probability regret bound* has mainly been considered (Abbasi-Yadkori et al., 2011; Agrawal and Goyal, 2013b), which bounds the pseudo-regret with high probability $1 - \delta$. Though such a bound can be transformed to an expected regret bound, this type of analysis often sacrifices the tightness since a linear regret might be suffered with small probability $\delta$. This is why the analysis in Vanchinathan et al. (2014) for BPM-LEAST finally yielded an $\tilde{O}(\sqrt{T})$ expected regret bound whereas their high-probability bound is $O(\log T)$.

## 4.1 Regret Upper Bound

In the following theorem, we show that logarithmic problem-dependent expected regret is achievable by the TSPM-Gaussian algorithm.

**Theorem 3** (Regret upper bound). *Consider any finite stochastic linear partial monitoring game. Assume that the game is strongly locally observable and $\Delta_i = (L_i - L_1)^\top p^* > 0$ for any $i \neq 1$. Then, the regret of TSPM-Gaussian satisfies for sufficiently large $T$ that*

$$\mathbb{E}[\mathrm{Reg}(T)] = O\left( \frac{AN^2 M \max_{i \in [N]} \Delta_i}{\Lambda^2} \log T \right), \tag{4}$$

*where $\Lambda := \min_{i \neq 1} \Lambda_i$ for $\Lambda_i = \Delta_i / \|z_{1,i}\|$ with $z_{1,i}$ defined after Definition 5.*

*Remark.* In the proof of Theorem 3, it is sufficient to assume that $L_1 - L_i \in \mathrm{Im}\, S_1^\top \oplus \mathrm{Im}\, S_i^\top$ for $i \in [N]$, which is weaker than the strong local observability, though it is still sometimes stronger than the local observability condition.

The proof of Theorem 3 is given in Appendix F. This result is the first problem-dependent bound of TS for PM, which also becomes the first logarithmic regret bound of TS for linear bandits.

The norm of $z_{j,k}$ in $\Lambda$ intuitively indicates the difficulty of the problem. Whereas we can estimate $(S_j p, S_k p)$ with noise through taking actions $j$ and $k$, the actual interest is the gap of the losses $p^\top (L_j - L_k) = (S_j p, S_k p)^\top z_{j,k}$. Thus, if $\|z_{j,k}\|$ is large, the gap estimation becomes difficult since the noise is enhanced through $z_{j,k}$.

Unfortunately, the derived bound in Theorem 3 has quadratic dependence on $N$, which seems to be not tight. This quadratic dependence comes from the difficulty of the *expected* regret analysis. In general, we evaluate the regret before and after the convergence of the statistics separately. Whereas the latter one usually becomes dominant, the main difficulty comes from the analysis of the former one, which might become large with low probability (Agrawal and Goyal, 2012; Kaufmann et al., 2012; Agrawal and Goyal, 2013a).

In our analysis, we were not able to bound the former one within a non-dominant order, though it is still logarithmic in $T$. In fact, our analysis shows that the regret after convergence is $\mathrm{O}(\sum_{i \neq 1} \Delta_i \frac{A}{\Lambda^2} \log T)$ as shown in Lemma 18 in Appendix F, which will become the regret with high probability. In particular, if we consider the classic bandit problem as a PM game, we can confirm that the derived bound after convergence becomes the best possible bound

$$\mathrm{O}\left(\sum_{i \neq 1} \frac{\log T}{\Delta_i}\right)$$

by considering $\Lambda_i$ depending on each suboptimal arm $i$ as the difficulty measure instead of $\Lambda$. Still, deriving a regret bound for the term before convergence within an non-dominant order is an important future work.

### 4.2 Technical Difficulties of the Analysis

The main difficulty of this regret analysis is that PM requires to consider the statistics of *all* actions when the number of selections $N_i(t)$ of some action $i$ is evaluated. This is in stark contrast to the analysis of the classic bandit problems, where it becomes sufficient to evaluate statistics of action $i$ and the best action 1. This makes the analysis remarkably complicated in TS, where we need to separately consider the randomness caused by the feedback and TS.

To overcome this difficulty, we handle the effect of actions of no interest in two different novel ways depending on each decomposed regret. The first one is to evaluate the worst-case effect of these actions based on an argument (Lemma 10) related to the law of the iterated logarithm (LIL), which is sometimes used in the best-arm identification literature to improve the performance (Jamieson et al., 2014). The second one is to bound the action-selection probability of TS using an argument of (super-)martingale (Theorem 16), which is of independent interest. Whereas such a technique is often used for the construction of confidence bounds (Abbasi-Yadkori et al., 2011), we reveal that it is also useful for evaluation of the regret of TS.

We only focused on the Gaussian noise $\epsilon_t \sim \mathcal{N}(0, I_M)$, rather than the more general sub-Gaussian noise. This restriction to the Gaussian noise comes from the essential difficulty of the problem-dependent analysis of TS, where lower bounds for some probabilities are needed whereas the sub-Gaussian assumption is suited for obtaining upper bounds. To the best of our knowledge, the problem-dependent regret analysis for TS on the sub-Gaussian case has never been investigated even for the multi-armed bandit setting, which is quite simple compared to that of PM. In the literature of the problem-dependent regret analysis, the noise distribution is restricted to distributions with explicitly given forms, e.g., Bernoulli, Gaussian, or more generally a one-dimensional canonical exponential family (Kaufmann et al., 2012; Agrawal and Goyal, 2013a; Korda et al., 2013). Their analysis relies on the specific characteristic of the distribution to bound the problem-dependent regret.

## 5 Experiments

In this section, we numerically compare the performance of TSPM and TSPM-Gaussian against existing methods, which are RandomPM (the algorithm which selects action randomly), FeedExp3 (Piccolboni and Schindelhauer, 2001), and BPM-TS (Vanchinathan et al., 2014). Recently, Lattimore and Szepesvári (2019) considered the sampling-based algorithm called Mario sampling for easy games. Mario sampling coincides with TS (except for the difference between pseudo-regret and regret with known time horizon) mentioned in the last section when any pair of actions is a neighbor. As shown in Appendix G, this property is indeed satisfied for dp-easy games defined in the following. Therefore, the performance is essentially the same between TSPM with $R = 1$ and Mario sampling. To compare the performance, we consider a dynamic pricing problem, which is a typical example of PM games. We conducted experiments on the discrete setting because the experiments for PM has been mainly focused on the discrete setting.

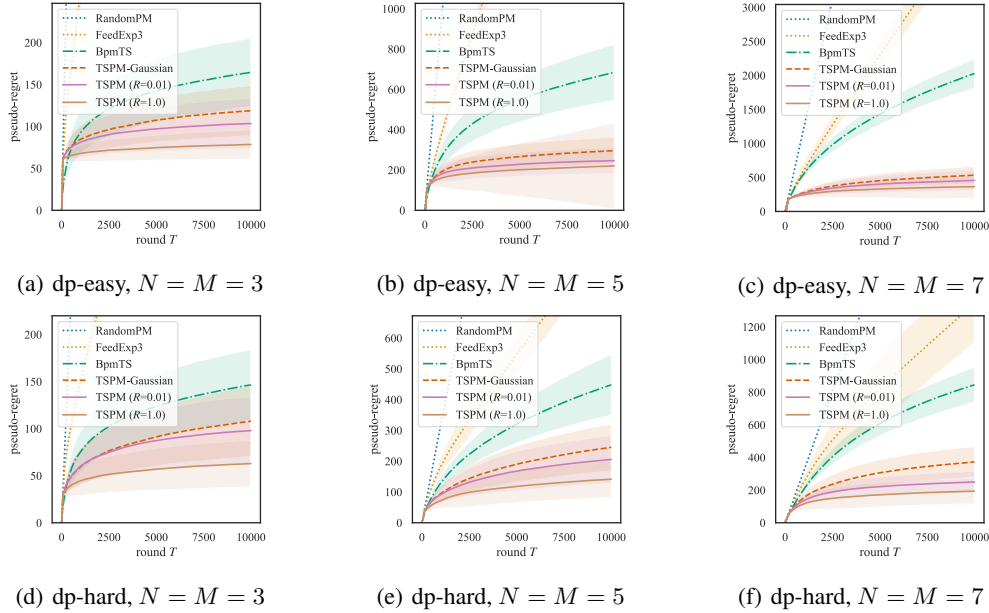

| | | |
|---|---|---|
| (a) dp-easy, $N = M = 3$ | (b) dp-easy, $N = M = 5$ | (c) dp-easy, $N = M = 7$ |
| (d) dp-hard, $N = M = 3$ | (e) dp-hard, $N = M = 5$ | (f) dp-hard, $N = M = 7$ |

**Figure 1:** Regret-round plots of algorithms. The solid lines indicate the average over 100 independent trials. The thin fillings are the standard error.

In the dynamic pricing game, the player corresponds to a seller, and the opponent corresponds to a buyer. At each round, the seller sells an item for a specific price $i(t)$, and the buyer comes with an evaluation price $j(t)$ for the item, where the selling price and the evaluation price correspond to the action and outcome, respectively. The buyer buys the item if the selling price $i(t)$ is smaller than or equal to $j(t)$ and not otherwise. The seller can only know if the buyer bought the item (denoted as feedback 0) or did not buy the item (denoted as 1). The seller aims to minimize the cumulative "loss", and there are two types of definitions for the loss, where each induced game falls into the easy and hard games. We call them *dp-easy* and *dp-hard* games, respectively.

In both cases, the seller incurs the constant loss $c > 0$ when the item is not bought due to the loss of opportunity to sell the item. In contrast, when the item is not bought, the loss incurred to the seller is different between these settings. The seller in the dp-easy game *does not* take the buyer's evaluation price into account. In other words, the seller gains the selling price $i(t)$ as a reward (equivalently incurs $-i(t)$ as a loss). Therefore, the loss for the selling price $i(t)$ and the evaluation $j(t)$ is

$$\ell_{i(t),j(t)} = -i(t)\mathbb{1}[i(t) \leq j(t)] + c\mathbb{1}[i(t) > j(t)].$$

This setting can be regarded as a generalized version of the online posted price mechanism, which was addressed in, *e.g.,* Blum et al. (2004) and Cesa-Bianchi et al. (2006), and an example of strongly locally observable games.

On the other hand, the seller in dp-hard game *does* take the buyer's evaluation price into account when the item is bought. In other words, the seller incurs the difference between the opponent evaluation and the selling price $j(t) - i(t)$ as a loss because the seller could have made more profit if the seller had sold at the price $j(t)$. Therefore, the loss incurred at time $t$ is

$$\ell_{i(t),j(t)} = (j(t) - i(t))\mathbb{1}[i(t) \leq j(t)] + c\mathbb{1}[i(t) > j(t)].$$

This setting is also addressed in Cesa-Bianchi et al. (2006), and belongs to the class of hard games. Note that our algorithm can also be applied to a hard game, though there is no theoretical guarantee.

**Setup.** In the both dp-easy and dp-hard games, we fixed $N = M \in \{3, 5, 7\}$ and $c = 2$. We fixed the time horizon $T$ to 10000 and simulated 100 times. For FeedExp3 and BPM-TS, the setup of hyperparameters follows their original papers. For TSPM, we set $\lambda = 0.001$, and $R$ was selected from $\{0.01, 1.0\}$. Here, recall that TSPM with $R = 1$ and $R = 0$ correspond to the exact sampling and TSPM-Gaussian, respectively, and a smaller value of $R$ gives the higher acceptance probability in the accept-reject sampling. Therefore, using small $R$ makes the algorithm time-efficient, although it can worsen the performance since it over-explores the tail of the posterior distributions. To stabilize

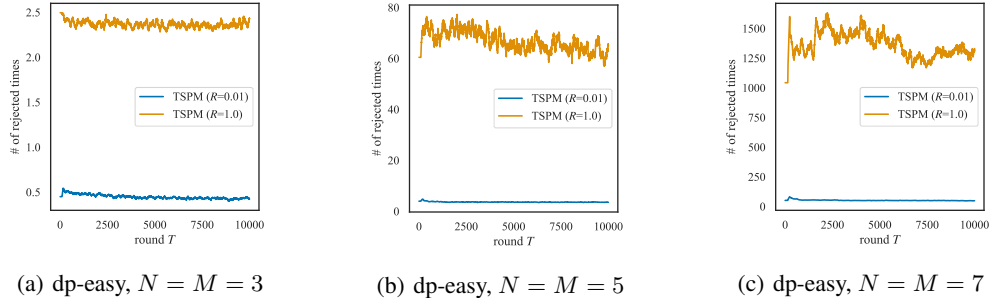

(a) dp-easy, $N = M = 3$        (b) dp-easy, $N = M = 5$        (c) dp-easy, $N = M = 7$

**Figure 2:** The number of rejected times by the accept-reject sampling. The solid lines indicate the average over 100 independent trials after taking moving average with window size 100.

sampling from the proposal distribution in Algorithm 3, we used an initialization that takes each action $n = 10A$ times. The detailed settings of the experiments with more results are given in Appendix H.

**Results.** Figure 1 is the empirical comparison of the proposed algorithms against the benchmark methods. This result shows that, in all cases, the TSPM with exact sampling gives the best performance. TSPM-Gaussian also outperforms BPM-TS even though both of them use Gaussian distributions as posteriors. Besides, the experimental results suggest that our algorithm performs reasonably well even for a hard game. It can be observed that the proposed methods outperform BPM-TS more significantly for a larger number of outcomes. Further discussion for this observation is given in Appendix D.

Figure 2 shows the number of rejections at each time step in the accept-reject sampling. We counted the number of times that either Line 4 in Algorithm 2 or Line 4 in Algorithm 3 was not satisfied. In the accept-reject sampling, it is desirable that the frequency of rejection does not increase as the time-step $t$ and does not increase rapidly with the number of outcomes. We can see that the former one is indeed satisfied. For the latter property, the frequency of rejection becomes unfortunately large when exact sampling ($R = 1$) is conducted. Still, we can substantially improve this frequency by setting $R$ to be a small value or zero, which still keeps regret tremendously better than that of BPM with almost the same time-efficiency as BPM-TS.

## 6    Conclusion and Discussion

This paper investigated Thompson sampling (TS) for stochastic partial monitoring from the algorithmic and theoretical viewpoints. We provided a new algorithm that enables exact sampling from the posterior distribution, and numerically showed that the proposed algorithm outperforms existing methods. Besides, we provided an upper bound for the problem-dependent logarithmic expected pseudo-regret for the linearized version of the partial monitoring. To our knowledge, this bound is the first logarithmic problem-dependent expected pseudo-regret bound of a TS-based algorithm for linear bandit problems and strongly locally observable partial monitoring games.

There are several remaining questions. As mentioned in Section 4, Kirschner et al. (2020) considered linear partial monitoring with the feedback structure $y(t) = S_{i(t)}p^* + \epsilon_t$, where $(\epsilon_t)_{t=1}^T$ is a sequence of independent sub-Gaussian noise vector in $\mathbb{R}^M$. This setting is the generalization of our linear setting, where $(\epsilon_t)_{t=1}^T$ are i.i.d. Gaussian vectors. Therefore, a natural question that arises is whether we can extend our analysis on TSPM-Gaussian to the sub-Gaussian case, although we believe it would be not straightforward as discussed in Section 4. It is also an important open problem to derive a regret bound on TSPM using the exact posterior sampling for the discrete partial monitoring. Although we conjecture that the algorithm also achieves logarithmic regret for the setting, there still remain some difficulties in the analysis. In particular, we have to handle the KL divergence in $f_t(p)$ and consider the restriction of the support of the opponent's strategy to $\mathcal{P}_M$, which make the analysis much more complicated. Besides, it is worth noting that the theoretical analysis of TS for hard games has never been theoretically investigated. We believe that in general TS suffers linear regret in the minimax sense due to its greediness. However, we conjecture that TS can achieve the sub-linear regret for some specific instances of hard games in the sense of the problem-dependent regret, as empirically observed in the experiments. Finally, it is an important open problem to derive the minimax regret for anytime TS-based algorithms. This needs more detailed analysis on $o(\log T)$ terms in the regret bound, which were dropped in our main result.

## Broader Impact

**Application.** Partial monitoring (PM) includes various online decision-making problems such as multi-armed bandits, linear bandits, dynamic pricing, and label efficient prediction. Not only can PM handles them, the dueling bandits, combinatorial bandits, transductive bandits, and many other problems can be seen as a partial monitoring game, as discussed in Kirschner et al. (2020). Therefore, our analysis of Thompson sampling (TS) for PM games pushes the application of TS to a more wide range of online decision-making problems forward. Moreover, PM has the potential that novel online-decision making problems are newly discovered, where we have to handle the limited feedback in an online fashion.

**Practical Use.** The obvious advantage of using TS is that the users can easily apply the algorithm to their problems. They do not have to solve mathematical optimization problems, which are often required to solve when using non-sampling-based algorithms (Bartók et al., 2012; Komiyama et al., 2015). For the negative side, the theoretical analysis for the regret upper bound might make the users become overconfident when the users use their algorithms. For example, they might use the TSPM algorithm to the linear PM game with heavy-tailed noise, such as sub-exponential noise, without noticing it. Nevertheless, this is not an TS-specific problem, but one that can be found in many theoretical studies, and TS is still one of the most promising policies.

## Acknowledgements

The authors would like to thank the meta-reviewer and reviewers for a lot of helpful comments. The authors would like to thank Kento Nozawa and Ikko Yamane for maintaining servers for our experiments, and Kenny Song for helpful discussion on the writing. TT was supported by Toyota-Dwango AI Scholarship, and RIKEN Junior Research Associate Program for the final part of the project. JH was supported by KAKENHI 18K17998, and MS was supported by KAKENHI 17H00757.

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
