[Supplementary Material]

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

# A Notation

Table 1 summarizes the symbols used in this paper.

**Table 1:** List of symbols used in this paper.

| Symbol | Meaning |
|---|---|
| $\mathcal{P}_n$ | $(n-1)$-dimensional probability simplex |
| $\|\cdot\|$ | Euclidian norm for vector and operator norm for matrix |
| $\|\cdot\|_p$ | $p$-norm |
| $\|\cdot\|_A$ | norm induced by positive semidefinite matrix $A$ |
| $\mathcal{D}_{\mathrm{KL}}\left(p\|q\right)$ | KL divergence from $q$ to $p$ |
| $B_r^n(p)$ | $n$-dimensional Euclidian ball of radius $r$ at point $p \in \mathbb{R}^N$ |
| $N, M \in \mathbb{N}$ | the number of actions and outcomes |
| $\Sigma$ | set of feedback symbols |
| $A$ | the number of feedback symbols |
| $p^* \in \mathcal{P}_M$ | opponent's strategy |
| $T$ | time horizon |
| $\mathbf{L} = (\ell_{i,j}) \in \mathbb{R}^{N \times M}$ | loss matrix |
| $\mathbf{H} = (h_{i,j}) \in \Sigma^{N \times M}$ | feedback matrix |
| $S_i \in \{0,1\}^{A \times M}\ (i=1,\dots,N)$ | signal matrix |
| $i(t)$ | action taken at time $t$ |
| $N_i(t)$ | the number of times the action $i$ is taken before time $t \in [T]$ |
| $j(t)$ | outcome taken by opponent at time $t$ |
| $y(t)$ | feedback observed at time $t$ |
| $F_t(p)$ | unnormalized posterior distribution in (1) |
| $f_t(p)$ | probability density function corresponding to $F_t(p)$ |
| $G_t(p)$ | unnormalized proposal distribution for $F_t(p)$ in (2) |
| $g_t(p)$ | probability density function corresponding to $G_t(p)$ |
| $q_i^{(t)} \in \mathcal{P}_M$ | empirical feedback distribution of action $i$ by time $t$ |
| $q_{i,n} \in \mathcal{P}_M$ | empirical feedback distribution of action $i$ after the action is taken $n$ times |
| $\mathcal{C}_i \subset \mathcal{P}_M$ | cell of action $i$ |

# B Posterior Distribution and Proposal Distribution in Section 3

In this appendix, we discuss representation of the posterior distribution and its relation with the proposal distribution.

**Proposition 4.** *$F_t(p)$ in (1) is proportional to the posterior distribution of the opponent's strategy, and $F_t(p) \leq G_t(p)$ for all $p \in \mathcal{P}_M$.*

*Proof.* The posterior distribution of the opponent's strategy parameter $\pi\left(p \mid \{i(s), y(s)\}_{s=1}^t\right)$ is rewritten as

$$
\begin{aligned}
\pi\left(p \mid \{i(s), y(s)\}_{s=1}^t\right) &\propto \pi\left(p, \{i(s), y(s)\}_{s=1}^t\right) \\
&\propto \pi(p) \prod_{s=1}^t \mathbb{P}\{y(s) \mid i(s), p\} \\
&= \pi(p) \prod_{i=1}^N \prod_{y=1}^A (S_{i,y}p)^{n_{iy}} \\
&\propto \pi(p) \prod_{i=1}^N \exp\left\{-n_i \mathcal{D}_{\mathrm{KL}}\left(q_i^{(t)} \middle\| S_i p\right)\right\},
\end{aligned}
\tag{5}
$$

where $S_{i,y}$ is the $i$-th row of the signal matrix $S_i$, and note that $q_i^{(t)}$ is the empirical feedback distribution of action $i$ at time $t$, that is, $q_i^{(t)} = [n_{i1}/n_i, \dots, n_{iA}/n_i]^\top \in \mathcal{P}_A$ for $n_{iy} = \sum_{s=1}^t \mathbb{1}[i(s)=i, y(s)=y]$ and $n_i = \sum_{y=1}^A n_{iy}$.

Next, we show that $F_t(p) \leq G_t(p)$ holds for all $p \in \mathcal{P}_M$. Using the Pinsker's inequality, the unnormalized posterior distribution $F_t(p)$ can be bounded from above as

$$
\begin{aligned}
F_t(p) &= \pi(p) \prod_{i=1}^N \exp\Big\{-n_i \mathcal{D}_{\mathrm{KL}}\Big(q_i^{(t)}\Big\|S_i p\Big)\Big\} \\
&\leq \pi(p) \prod_{i=1}^N \exp\Big\{-\frac{1}{2}n_i\|q_i^{(t)} - S_i p\|_1^2\Big\} \quad \text{(by Pinsker's ineq.)} \\
&= \pi(p) \exp\Big\{-\frac{1}{2}\sum_{i=1}^N n_i\|q_i^{(t)} - S_i p\|_1^2\Big\} \\
&\leq \pi(p) \exp\Big\{-\frac{1}{2}\sum_{i=1}^N n_i\|q_i^{(t)} - S_i p\|^2\Big\} \quad \Big(\text{by } \|q_i^{(t)} - S_i p\|_1 \geq \|q_i^{(t)} - S_i p\|\Big) \\
&= G_t(p).
\end{aligned}
\tag{6}
$$

$\square$

*Remark.* The unnormalized density $G_t(p)$ is indeed Gaussian. Recalling that $B_t$ and $b_t$ are defined in (3) as

$$
B_t = \sum_{i=1}^N n_i S_i^\top S_i = \sum_{s=1}^t S_{i(s)}^\top S_{i(s)} = B_{t-1} + S_{i(t)}^\top S_{i(t)}, \quad b_t = \sum_{i=1}^N n_i S_i^\top q_i^{(t)} = b_{t-1} + S_{i(t)}^\top e_{y(t)},
\tag{7}
$$

we have

$$
\begin{aligned}
\sum_{i=1}^N n_i\|q_i^{(t)} - S_i p\|^2 &= \sum_{i=1}^N n_i(q_i^{(t)} - S_i p)^\top (q_i^{(t)} - S_i p) \\
&= p^\top \underbrace{\Big(\sum_{i=1}^N n_i S_i^\top S_i\Big)}_{B_t} p, -2 \underbrace{\Big(\sum_{i=1}^N n_i S_i^\top q_i^{(t)}\Big)^\top}_{b_t} p + \underbrace{\sum_{i=1}^N n_i\|q_i^{(t)}\|^2}_{c_t} \\
&= p^\top B_t p - 2b_t^\top p + c_t \\
&= (p - B_t^{-1}b_t)^\top B_t(p - B_t^{-1}b_t) + c_t - b_t^\top B_t^{-1}b_t.
\end{aligned}
\tag{8}
$$

Therefore, we have

$$
\exp\Big\{-\frac{1}{2}\sum_{i=1}^N n_i\|q_i^{(t)} - S_i p\|^2\Big\} \propto \exp\Big\{-\frac{1}{2}(p - B_t^{-1}b_t)^\top B_t(p - B_t^{-1}b_t)\Big\}.
\tag{9}
$$

## C  Proof of Proposition 2

We will see that the the procedure sampling $\tilde{p}_t$ from $g_t(p)$ and Algorithm 3 are equivalent. First, we derive the Gaussian density of $g_t(p)$ projected onto $\{p \in \mathbb{R}^M : \sum_{i=1}^M p_i = 1\}$.

For simplicity, we omit the subscript $t$ and write, *e.g.,* $B$ instead of $B_t$. We define $p = [p^{(\alpha)\top}, p_M]^\top \in \mathbb{R}^{M-1} \times \mathbb{R}$. Let $h = B^{-1}b$, and define $h = [h^{(\alpha)\top}, h_M]^\top \in \mathbb{R}^{M-1} \times \mathbb{R}$. Let $B = \begin{bmatrix} C & d \\ d^\top & f \end{bmatrix}$, where $C \in \mathbb{R}^{M-1 \times M-1}, d \in \mathbb{R}^{M-1}$, and $f \in \mathbb{R}$. Also, let $b = [b^{(\alpha)\top}, b^{(M)}]^\top \in \mathbb{R}^{M-1} \times \mathbb{R}$.

Using the decomposition

$$
(p - B^{-1}b)^\top B(p - B^{-1}b) = \underbrace{p^\top B p}_{(a)} - 2\underbrace{h^\top B p}_{(b)} + h^\top B h,
\tag{10}
$$

we rewrite each term by restricting the domain of $p$ so that it satisfies the condition $\sum_{i=1}^{M} p_i = 1$. Now the first term (a) is rewritten as

$$(a) = {p^{(\alpha)}}^\top C p^{(\alpha)} + 2{p^{(\alpha)}}^\top d p_M + f p_M^2$$

$$= {p^{(\alpha)}}^\top C p^{(\alpha)} + \underbrace{2\,{p^{(\alpha)}}^\top d\Big(1 - \sum_{i=1}^{M-1} p_i\Big)}_{(a1)} + \underbrace{f\Big(1 - \sum_{i=1}^{M-1} p_i\Big)^2}_{(a2)}. \tag{11}$$

The term (a1) is rewritten as

$$(a1) = {p^{(\alpha)}}^\top d - {p^{(\alpha)}}^\top d \sum_{i=1}^{M-1} p_i$$

$$= {p^{(\alpha)}}^\top d - {p^{(\alpha)}}^\top d \mathbf{1}_{M-1}^\top p^{(\alpha)}$$

$$= {p^{(\alpha)}}^\top d - {p^{(\alpha)}}^\top D p^{(\alpha)} \quad \Big(D = \frac{1}{2}\big(d\mathbf{1}_{M-1}^\top + \mathbf{1}_{M-1} d^\top\big)\Big), \tag{12}$$

and the term (a2) is rewritten as

$$(a2) = \Big(1 - \sum_{i=1}^{M-1} p_i\Big)^2$$

$$= 1 - 2\sum_{i=1}^{M-1} p_i + \Big(\sum_{i=1}^{M-1} p_i\Big)^2$$

$$= 1 - 2\mathbf{1}_{M-1}^\top p^{(\alpha)} + {p^{(\alpha)}}^\top \mathbf{1}_{M-1}\mathbf{1}_{M-1}^\top p^{(\alpha)}. \tag{13}$$

Therefore,

$$(a) = {p^{(\alpha)}}^\top \underbrace{(C - 2D + f\mathbf{1}_{M-1}\mathbf{1}_{M-1}^\top)}_{\tilde{B}} p^{(\alpha)} - 2(f\mathbf{1}_{M-1} - d)^\top p^{(\alpha)} + f. \tag{14}$$

With regard to the term (b), we have

$$(b) = b^\top p$$

$$= {b^{(\alpha)}}^\top {p^{(\alpha)}}^\top + b^{(M)} p_M$$

$$= (b^{(\alpha)} - b^{(M)}\mathbf{1}_{M-1})^\top p^{(\alpha)} + b^{(M)}. \tag{15}$$

Therefore,

$$(p - B^{-1}b)^\top B(p - B^{-1}b)$$

$$= {p^{(\alpha)}}^\top \tilde{B} p^{(\alpha)} - 2(\underbrace{f\mathbf{1}_{M-1} - d + b^{(\alpha)} - b^{(M)}\mathbf{1}_{M-1}}_{\tilde{b}})^\top p^{(\alpha)} + f - 2b^{(M)} + h^\top B h$$

$$= (p^{(\alpha)} - \tilde{B}^{-1}\tilde{b})^\top \tilde{B}(p^{(\alpha)} - \tilde{B}^{-1}\tilde{b}) + f - 2b^{(M)} - \tilde{b}^\top \tilde{B}^{-1}\tilde{b} + b^\top B^{-1}b \quad \big(\text{by } h^\top B h = b^\top B^{-1}b\big). \tag{16}$$

From the above argument, the density $\mathcal{N}(\tilde{B}^{-1}b,\ \tilde{B}^{-1})$ is the Gaussian distribution of $g_t(p)$ on $\{p \in \mathbb{R}^M : \sum_{i=1}^{M} p_i = 1\}$. Therefore, the $p = [{p^{(\alpha)}}^\top, 1 - \sum_{i=1}^{M-1}(p^{(\alpha)})_i]^\top$ for $p^{(\alpha)} \sim \mathcal{N}(\tilde{B}^{-1}b,\ \tilde{B}^{-1})$ is supported over $\{p \in \mathbb{R}^M : \sum_{i=1}^{M} p_i = 1\}$.

If the sample $p^{(\alpha)}$ from $\mathcal{N}(\tilde{B}^{-1}b,\ \tilde{B}^{-1})$ is in $\mathcal{P}_{M-1}$, then we can obtain the last element $p^{(M)}$ by $p^{(M)} = 1 - \sum_{i=1}^{M-1}(p^{(\alpha)})_i$. Otherwise, the probability that $p^{(\alpha)}$ is the first $M - 1$ elements of the sample from $g_t(p)$ is zero, and hence, $[{p^{(\alpha)}}^\top, p^{(M)}]^\top$ cannot be a sample from $g_t(p)$. Therefore, sampling $\tilde{p}_t$ from $g_t(p)$ and Algorithm 3 are equivalent.

# D  Relation between TSPM-Gaussian and BPM-TS

In this appendix, we discuss the relation between TSPM-Gaussian and BPM-TS (Vanchinathan et al., 2014).

**Underlying Feedback Structure.**  Here, we discuss the underlying feedback structure behind TSPM-Gaussian and BPM-TS.

We first consider the underlying feedback structure behind BPM-TS. In the following, we see that the feedback structure

$$y(t) = S_{i(t)}p + S_{i(t)}\epsilon \,,\ \epsilon \sim \mathcal{N}(0, I_M) \tag{17}$$

induces the posterior distribution in BPM-TS. Under this feedback structure, we have $y(t) \sim \mathcal{N}(S_{i(t)}p, S_{i(t)}S_{i(t)}^\top)$.

When we take the prior distribution $\pi(p)$ as $\mathcal{N}(0, \sigma_0^2 I_M)$, the posterior distribution for the opponent's strategy parameter can be written as

$$\pi\Big(p \;\Big|\; \{i(s), y(s)\}_{s=1}^t\Big)$$

$$\propto \pi(p) \prod_{s=1}^t \pi(y(s) \mid i(s),\, p)$$

$$= \pi(p) \prod_{s=1}^t \mathbb{P}_{y \sim \mathcal{N}(S_{i(s)}p, S_{i(s)}S_{i(s)}^\top)} \{y = y(s)\}$$

$$= \exp\Big(-\frac{p^\top p}{2\sigma_0^2}\Big) \prod_{s=1}^t \exp\Big(-\frac{1}{2}(y(s) - S_{i(s)}p)^\top (S_{i(s)}S_{i(s)}^\top)^{-1}(y(s) - S_{i(s)}p)\Big)$$

$$= \exp\Big\{-\frac{1}{2}\Big(p^\top \Big(\frac{1}{\sigma_0^2}I_M + \sum_{s=1}^T S_{i(s)}^\top (S_{i(s)}S_{i(s)}^\top)^{-1} S_{i(s)}\Big)p\Big)$$

$$\qquad - 2\Big(\sum_{s=1}^t y(s)^\top (S_{i(s)}S_{i(s)}^\top)^{-1} S_{i(s)}p\Big) + \text{(a term independent of } p)\Big\}$$

$$\propto \exp\Big\{-\frac{1}{2}(p^\top B_t^{\mathrm{BPM}}p - 2b_t^{\mathrm{BPM}\top}p)\Big\}$$

$$\propto \exp\Big\{-\frac{1}{2}(p - B_t^{\mathrm{BPM}^{-1}}b_t^{\mathrm{BPM}})^\top B_t^{\mathrm{BPM}}(p - B_t^{\mathrm{BPM}^{-1}}b_t^{\mathrm{BPM}})\Big\} \,, \tag{18}$$

where

$$B_t^{\mathrm{BPM}} = \frac{1}{\sigma_0^2}I_M + \sum_{s=1}^t S_{i(s)}^\top (S_{i(s)}S_{i(s)}^\top)^{-1} S_{i(s)} = B_{t-1}^{\mathrm{BPM}} + S_{i(t)}^\top (S_{i(t)}S_{i(t)}^\top)^{-1} S_{i(t)} \,, \tag{19}$$

$$b_t^{\mathrm{BPM}} = \sum_{s=1}^t S_{i(s)}^\top (S_{i(s)}S_{i(s)}^\top)^{-1} y(s) = b_{t-1}^{\mathrm{BPM}} + S_{i(t)}^\top (S_{i(t)}S_{i(t)}^\top)^{-1} y(t) \,. \tag{20}$$

Therefore, the posterior distribution $\pi\Big(p \mid \{i(s), y(s)\}_{s=1}^t\Big)$ is

$$\frac{1}{\sqrt{(2\pi)^M |B_t^{\mathrm{BPM}^{-1}}|}} \exp\Big\{-\frac{1}{2}(p - B_t^{\mathrm{BPM}^{-1}}b_t^{\mathrm{BPM}})^\top B_t^{\mathrm{BPM}}(p - B_t^{\mathrm{BPM}^{-1}}b_t^{\mathrm{BPM}})\Big\} \,. \tag{21}$$

and this distribution indeed corresponds to the posterior distribution in BPM-TS (Vanchinathan et al., 2014) with $B_t^{\mathrm{BPM}} = \Sigma_t^{-1}$.

Using the same argument, we can confirm that the feedback structure

$$y_t = S_i p + \epsilon \,,\ \epsilon \sim \mathcal{N}(0, I_M) \,. \tag{22}$$

induces

$$\bar{g}_t(p) := \frac{1}{\sqrt{(2\pi)^M |B_t^{-1}|}} \exp\left(-\frac{1}{2}\left\|p - B_t^{-1}b_t\right\|_{B_t}^2\right), \tag{23}$$

which corresponds to the posterior distribution for TSPM in linear partial monitoring.

**Covariances in TSPM-Gaussian and BPM-TS.** In the linear partial monitoring, TSPM assumes noise with covariance $I_M$, which is compatible with the fact that the discrete setting can be regarded as linear PM with $I_M$-sub-Gaussian noise. On the other hand, BPM-TS assumes covariance $S_i S_i^\top$, and in general $I_M \preceq S_i S_i^\top$ holds. Therefore, BPM-TS assumes unnecessarily larger covariance, which makes learning slow down.

# E   Preliminaries for Regret Analysis

In this appendix, we give some technical lemmas, which are used for the derivation of the regret bound in Appendix F. Here, we write $X \succeq Y$ to denote $X - Y \succeq 0$. For $a, b \in \mathbb{R}$, let $a \wedge b$ be $a$ if $a \leq b$ otherwise $b$, and $a \vee b$ be $b$ if $a \leq b$ otherwise $a$. We use $h(a) := \mathbb{P}_{X \sim \chi_M^2}\{X \geq a\}$ to evaluate the behavior of the posterior samples, where $\chi_M^2$ is the chi-squared distribution with $M$ degree of freedom.

## E.1   Basic Lemmas

**Fact 5** (Moment generating function of squared-Gaussian distribution). *Let $X$ be the random variable following the standard normal distribution. Then, the moment generating function of $X^2$ is* $\mathbb{E}\left[\exp(\xi X^2)\right] = (1 - 2\xi)^{-1/2}$ *for $\xi < 1/2$.*

**Lemma 6** (Chernoff bound for chi-squared random variable). *Let $X$ be the random variable following the chi-squared distribution with $k$ degree of freedom. Then, for any $a \geq 0$ and $0 \leq \xi < 1/2$,*

$$\mathbb{P}\{X \geq a\} \leq e^{-\xi a}(1 - 2\xi)^{-\frac{k}{2}}. \tag{24}$$

*Proof.* By Markov's inequality, the LHS can be bounded as

$$
\begin{aligned}
\mathbb{P}\{X \geq a\} &= \mathbb{P}\left\{\sum_{i=1}^{k} X_i^2 \geq a\right\} \quad (X_1, \ldots, X_k \overset{\text{i.i.d.}}{\sim} \mathcal{N}(0,1)) \\
&= \mathbb{P}\left\{\exp\left(\xi \sum_{i=1}^{k} X_i^2\right) \geq \exp(\xi a)\right\} \\
&\leq e^{-\xi a}\left(\mathbb{E}\left[e^{\xi X_1^2}\right]\right)^k \quad \text{(by Markov's ineq.)} \\
&= e^{-\xi a}(1 - 2\xi)^{-\frac{k}{2}} \quad \text{(by Fact 5)} ,
\end{aligned}
\tag{25}
$$

which completes the proof. $\qquad\square$

## E.2   Property of Strong Local Observability

Recall that $\Delta_i = (L_i - L_1)^\top p^* > 0$ for $i \in [N]$, which is the difference of the expected loss of actions $i$ and $1$. For this define

$$\epsilon := \left(\frac{1}{2\sqrt{A}} \min_{i \neq 1} \frac{\Delta_i}{\|z_{1,i}\|}\right) \wedge \left(\min_{p \in \mathcal{C}_1^c} \frac{4}{3}\|p - p^*\|\right) , \tag{26}$$

which is used throughout the proof of this appendix and Appendix F. The following lemma provides the key property of the strong local observability condition.

**Lemma 7.** *For any partial monitoring game with strong local observability and $p \in \mathbb{R}^M$, any of the conditions 1–3 in the following is not satisfied:*

1. *$L_1^\top p > L_k^\top p$   (Worse action $k$ looks better under $p$.)*

2. *$\|S_1 p - S_1 p^*\| \leq \epsilon$*

3. $\|S_k p - S_k p^*\| \leq \epsilon$ .

*Proof.* We prove by contradiction. Assume that there exists $p \in \mathbb{R}^M$ such that conditions 1–3 are simultaneously satisfied.

Now, by the conditions 2 and 3, we have

$$\begin{aligned} |S_1 p - S_1 p^*| &\preceq \epsilon \mathbf{1}_A \,, \\ |S_k p - S_k p^*| &\preceq \epsilon \mathbf{1}_A \,. \end{aligned} \tag{27}$$

Here, $|\cdot|$ is the element-wise absolute value, and $\preceq$ means that the inequality $\leq$ holds for each element. Therefore,

$$\left| \begin{pmatrix} S_1 \\ S_k \end{pmatrix} (p - p^*) \right| \preceq \epsilon \mathbf{1}_{2A} \,. \tag{28}$$

On the other hand, by the strong local observability condition, for any $k \neq 1$, there exists $z_{1,k} \neq 0 \in \mathbb{R}^{2A}$ such that

$$(L_1 - L_k)^\top = z_{1,k}^\top \begin{pmatrix} S_1 \\ S_k \end{pmatrix} \,. \tag{29}$$

Now, we have

$$\begin{aligned} &z_{1,k}^\top \begin{pmatrix} S_1 \\ S_k \end{pmatrix} (p - p^*) \\ &\leq \|z_{1,k}\| \left\| \begin{pmatrix} S_1 \\ S_k \end{pmatrix} (p - p^*) \right\| \quad \text{(by Cauchy-Schwarz ineq.)} \\ &\leq \sqrt{2A} \epsilon \|z_{1,k}\| \quad \text{(by Eq. (28))} \,, \end{aligned} \tag{30}$$

and

$$\begin{aligned} &z_{1,k}^\top \begin{pmatrix} S_1 \\ S_k \end{pmatrix} (p - p^*) \\ &= (L_1 - L_k)^\top (p - p^*) \quad \text{(by Eq. (29))} \\ &= (L_1 - L_k)^\top p + (L_k - L_1)^\top p^* \\ &\geq \Delta_k \quad \text{(by Condition 1 \& def. of } \Delta_k) \,. \end{aligned} \tag{31}$$

Therefore, from (30) and (31), we have

$$\Delta_k \leq \sqrt{2A} \epsilon \|z_{1,k}\| \,. \tag{32}$$

This inequality does not hold for all $k \neq 1$ for the predefined value of $\epsilon$, since we have

$$\epsilon \leq \frac{1}{2\sqrt{A}} \min_{k \neq 1} \frac{\Delta_k}{\|z_{1,k}\|} \,. \tag{33}$$

Therefore, the proof is completed by contradiction. □

*Remark.* The similar result holds when the optimal action 1 is replaced with action $j \neq k$ such that $\Delta_{j,k} := (L_j - L_k)^\top p^* > 0$ by taking $\epsilon$ satisfying

$$\epsilon \leq \frac{1}{2\sqrt{A}} \min_{j \neq k : \Delta_{j,k} > 0} \frac{\Delta_{j,k}}{\|z_{j,k}\|} \,. \tag{34}$$

From Lemma 7, we have the following corollary.

**Corollary 8.** *For any $p \in \mathbb{R}^M$ satisfying $p \in \mathcal{C}_i$ and $\|S_1 p - S_1 p^*\| \leq \epsilon$, we have*

$$\|S_i p - S_i p^*\| > \epsilon \,. \tag{35}$$

*Proof.* Note that $p \in \mathcal{C}_i$ is equivalent to $(L_1 - L_i)^\top p^* > 0$ for any $i \neq 1$. Therefore, the result directly follows from Lemma 7. □

The next lemma is the property of Mahalanobis distance corresponding to $\bar{g}_t(p)$.

**Lemma 9.** *Define* $\mathcal{T}_i := \{p \in \mathbb{R}^M : \|S_i p - S_i p^*\| > \epsilon\}$. *Assume that* $N_i(t) \geq n_i$, $\|S_i \hat{p}_t - S_i p^*\| \leq \epsilon/4$. *Then, for any* $0 \leq \xi < 1/2$

$$h\left(\inf_{p \in \mathcal{T}_i} \|B_t^{1/2}(p - \hat{p}_t)\|^2\right) \leq \exp\left(-\frac{9}{16}\xi\epsilon^2 n_i\right)(1 - 2\xi)^{-M/2}. \tag{36}$$

*Proof.* To bound the LHS of the above inequality, we bound $\|B_t^{1/2}(p - \hat{p}_t)\|^2$ from below for $p \in \mathcal{T}_i$. Using the triangle inequality and the assumptions, we have

$$\begin{aligned}\|S_i(p - \hat{p}_t)\| &\geq \|S_i p - S_i p^*\| - \|S_i \hat{p}_t - S_i p^*\| \\ &> \epsilon - \epsilon/4 > 0.\end{aligned} \tag{37}$$

Therefore, we have

$$\begin{aligned}\|B_t^{1/2}(p - \hat{p}_t)\|^2 &\geq \sum_{k \in [N]} N_k(t)\|S_k(p - \hat{p}_t)\|^2 \quad \text{(by def. of } B_t) \\ &\geq n_i \|S_i(p - \hat{p}_t)\|^2 \quad (N_i(t) \geq n_i) \\ &> \frac{9}{16}\epsilon^2 n_i \quad \text{(by Eq. (37))}.\end{aligned} \tag{38}$$

By the Chernoff bound for a chi-squared random variable in Lemma 6, we now have

$$h(a) \leq e^{-\xi a}(1 - 2\xi)^{-M/2}, \tag{39}$$

for any $a \geq 0$ and $0 \leq \xi < 1/2$. Hence, using the fact that $\|B_t^{1/2}(p - \hat{p}_t)\|^2$ follows the chi-squared distribution with $M$ degree of freedom, we have

$$\begin{aligned}h\left(\inf_{p \in \mathcal{T}_i} \|B_t^{1/2}(p - \hat{p}_t)\|^2\right) &\leq h\left(\frac{9}{16}\epsilon^2 n_i\right) \\ &\leq \exp\left(-\frac{9}{16}\xi\epsilon^2 n_i\right)(1 - 2\xi)^{-M/2},\end{aligned} \tag{40}$$

which completes the proof. $\square$

### E.3 Statistics of Uninterested Actions

For any $k \neq i$ and $n_k \in [T]$, define

$$Z_{n_k} := n_k \|q_{k,n_k} - S_k p^*\|^2, \tag{41}$$

$$Z_{\backslash i} := \sum_{k \neq i} \max_{n_k \in [T]} Z_{n_k}. \tag{42}$$

In this section, we bound $\mathbb{E}\left[Z_{\backslash i}\right]$ from above. Note that $Z_{\backslash i}$ is independent of the randomness of Thompson sampling.

**Lemma 10** (Upper bound for the expectation of $Z_{\backslash i}$)**.**

$$\mathbb{E}\left[Z_{\backslash i}\right] \leq 4N\left(\log T + \frac{A}{2}\log 2 + 1\right). \tag{43}$$

*Proof.* Recall that in linear partial monitoring, the feedback $y(t) \in \mathbb{R}^A$ for action $k$ is given as

$$y_t = S_k p^* + \epsilon, \ \epsilon \sim \mathcal{N}(0, I_A) \tag{44}$$

at round $t \in [T]$, Therefore, $y(t) - S_k p^* \sim \mathcal{N}(0, I_A)$. Since $q_{k,n_k} = \frac{1}{n_k}\sum_{s \in [T]:i(s)=k} y(s)$ for any $n_k \in [T]$, we have

$$q_{k,n_k} - S_k p^* = \frac{1}{n_k}\sum_{s \in [T]:i(s)=k}(y(s) - S_k p^*) \sim \mathcal{N}(0, I_A/n_k). \tag{45}$$

Therefore,

$$\sqrt{n_k}(q_{k,n_k} - S_k p^*) \sim \mathcal{N}(0, I_A)\,, \tag{46}$$

and thus

$$n_k \| q_{k,n_k} - S_k p^* \|^2 = \| \sqrt{n_k}(q_{k,n_k} - S_k p^*) \|^2 \sim \chi_A^2\,. \tag{47}$$

Therefore, for any $0 \le \xi < 1/2$,

$$
\begin{aligned}
\mathbb{E}\left[ \max_{n_k \in [T]} Z_{n_k} \right] &= \int_0^\infty \mathbb{P}\left\{ \max_{n_k \in [T]} Z_{n_k} \ge x \right\} \mathrm{d}x \\
&\le \int_0^\infty \left[ 1 \wedge T \cdot \mathbb{P}\{ Z_1 \ge x \} \right] \mathrm{d}x \quad \text{(by the union bound)} \\
&\le \int_0^\infty \left[ 1 \wedge T \cdot \mathrm{e}^{-\xi x}(1 - 2\xi)^{-\frac{A}{2}} \right] \mathrm{d}x \quad \left( \text{by } Z_1 \sim \chi_{(}^2 A) \text{ and Lemma } 6 \right) \\
&= \int_0^{x^*} \mathrm{d}x + \int_{x^*}^\infty T \cdot \mathrm{e}^{-\xi x}(1 - 2\xi)^{-\frac{A}{2}} \mathrm{d}x \\
&\le x^* + T \cdot \int_{x^*}^\infty \mathrm{e}^{-\xi x}(1 - 2\xi)^{-\frac{A}{2}} \mathrm{d}x \\
&= x^* + T(1 - 2\xi)^{-\frac{A}{2}} \left[ -\frac{1}{\xi} \mathrm{e}^{-\xi x} \right]_{x^*}^\infty \\
&= \frac{1}{\xi}\left\{ \log T - \frac{A}{2} \log(1 - 2\xi) + 1 \right\}, \tag{48}
\end{aligned}
$$

where $x^* := \frac{1}{\xi}\left\{ \log T - \frac{A}{2} \log(1 - 2\xi) \right\}$. Therefore, taking $\xi = 1/4$, we have

$$
\begin{aligned}
\mathbb{E}\left[ Z_{\setminus i} \right] = \mathbb{E}\left[ \sum_{k \ne i} \max_{n_k \in [T]} Z_{n_k} \right] \\
&\le \sum_{k \ne i} \mathbb{E}\left[ \max_{n_k \in [T]} Z_{n_k} \right] \\
&\le (N - 1)\frac{1}{\xi}\left\{ \log T - \frac{A}{2} \log(1 - 2\xi) + 1 \right\} \\
&\le 4N\left( \log T + \frac{A}{2} \log 2 + 1 \right), \tag{49}
\end{aligned}
$$

which completes the proof. $\qquad\square$

### E.4 Mahalanobis Distance Process

Discussions in this section are essentially very similar to Abbasi-Yadkori et al. (2011, Lemma 11), but their results are not directly applicable and we give the full derivation for self-containedness. To maximize the applicability here we only assume sub-Gaussian noise rather than a Gaussian one.

Let $\epsilon_t$ be zero-mean 1-sub-Gaussian random variable, which satisfies

$$\mathbb{E}\left[ \mathrm{e}^{\lambda^\top \epsilon_t} \right] \le \mathrm{e}^{-\frac{\|\lambda\|^2}{2}} \tag{50}$$

for any $\lambda \in \mathbb{R}^M$.

**Lemma 11.** *For any vector $v \in \mathbb{R}^M$ and positive definite matrix $V \in \mathbb{R}^{M \times M}$ such that $V \succ I$,*

$$\mathbb{E}_{\epsilon_t}\left[ \mathrm{e}^{\frac{\|\epsilon_t + v\|_{V^{-1}}^2}{2}} \right] \le \frac{\sqrt{|V|}}{\sqrt{|V - I|}} \mathrm{e}^{\frac{1}{2} v^\top (V - I)^{-1} v}\,. \tag{51}$$

*Proof.* For any $x \in \mathbb{R}^M$

$$\mathbb{E}_{\lambda \sim \mathcal{N}(0, V^{-1})}\left[ \mathrm{e}^{\lambda^\top x} \right] = \mathrm{e}^{\frac{\|x\|_{V^{-1}}^2}{2}}\,. \tag{52}$$

Therefore, by letting $x = \epsilon_t + v$ we see that

$$\mathrm{e}^{\frac{\|\epsilon_t + v\|^2_{V^{-1}}}{2}} = \mathbb{E}_{\lambda \sim \mathcal{N}(0, V^{-1})}\left[\mathrm{e}^{\lambda^\top(\epsilon_t + v)}\right]. \tag{53}$$

As a result, by the definition of sub-Gaussian random variables, we have

$$\begin{aligned}
\mathbb{E}_{\epsilon_t}\left[\mathrm{e}^{\frac{\|\epsilon_t + v\|^2_{V^{-1}}}{2}}\right] &= \mathbb{E}_{\lambda \sim \mathcal{N}(0, V^{-1})}\left[\mathbb{E}_{\epsilon_t}\left[\mathrm{e}^{\lambda^\top(\epsilon_t + v)}\right]\right] \\
&= \mathbb{E}_{\lambda \sim \mathcal{N}(0, V^{-1})}\left[\mathrm{e}^{\lambda^\top v}\mathbb{E}_{\epsilon_t}\left[\mathrm{e}^{\lambda^\top \epsilon_t}\right]\right] \\
&\leq \mathbb{E}_{\lambda \sim \mathcal{N}(0, V^{-1})}\left[\mathrm{e}^{\lambda^\top v}\mathrm{e}^{\|\lambda\|^2/2}\right] \\
&= \frac{1}{(2\pi)^{d/2}\sqrt{|V^{-1}|}}\int \mathrm{e}^{\lambda^\top v}\mathrm{e}^{\|\lambda\|^2/2}\mathrm{e}^{-\|\lambda\|^2_V/2}\mathrm{d}\lambda \\
&= \frac{1}{(2\pi)^{d/2}\sqrt{|V^{-1}|}}\int \mathrm{e}^{-\frac{1}{2}\left(\lambda^\top(V-I)\lambda - 2v^\top\lambda\right)}\mathrm{d}\lambda \\
&= \frac{\sqrt{|V-I|}}{(2\pi)^{d/2}\sqrt{|V^{-1}||V-I|}}\int \mathrm{e}^{-\frac{1}{2}\left((\lambda-(V-I)^{-1}v)^\top(V-I)(\lambda-(V-I)^{-1}v) - v^\top(V-I)^{-1}v\right)}\mathrm{d}\lambda \\
&= \frac{\sqrt{|V|}}{\sqrt{|V-I|}}\mathrm{e}^{\frac{1}{2}v^\top(V-I)^{-1}v}. \tag{54}
\end{aligned}$$

$\square$

**Lemma 12.**

$$\mathbb{E}\left[\exp\left(\frac{1}{2}\left(\|\hat{p}_t - p^*\|^2_{B_t} - \|\hat{p}_{t-1} - p^*\|^2_{B_{t-1}}\right)\right) \,\middle|\, \hat{p}_{t-1}, B_{t-1}, S_{i(t-1)}\right] \leq \sqrt{\frac{|B_t|}{|B_{t-1}|}}. \tag{55}$$

*Proof.* Let $Z_t := -\lambda p^* + \sum_{s=1}^t S_{i(s)}^\top \epsilon_s$, and we have

- $B_t = \lambda I + \sum_{s=1}^t S_{i(s)}^\top S_{i(s)}$,

- $b_t = \sum_{s=1}^t S_{i(s)}^\top y(s) = B_t p^* + Z_t$,

- $\hat{p}_t = B_t^{-1}b_t = p^* + B_t^{-1}Z_t$.

In the following we omit the conditioning on $(\hat{p}_{t-1}, B_{t-1}, S_{i(t-1)})$ for notational simplicity.

Let us define $C_t := S_{i(t)}B_{t-1}S_{i(t)}^\top$ and $d_t := S_{i(t)}B_{t-1}^{-1}Z_{t-1} = S_{i(t)}(\hat{p}_t - p^*)$. Then, using the Sherman-Morrison-Woodbury formula we have

$$\|\hat{p}_t - p^*\|^2_{B_t} - \|\hat{p}_{t-1} - p^*\|^2_{B_{t-1}}$$

$$= Z_t^\top B_t^{-1} Z_t - Z_{t-1}^\top B_{t-1}^{-1} Z_{t-1}$$

$$= (Z_{t-1}^\top + \epsilon_t^\top S_{i(t)})(B_{t-1}^{-1} - B_{t-1}^{-1} S_{i(t)}^\top (I + S_{i(t)} B_{t-1}^{-1} S_{i(t)}^\top)^{-1} S_{i(t)} B_{t-1}^{-1})(Z_{t-1} + S_{i(t)}^\top \epsilon_t) - Z_{t-1}^\top B_{t-1}^{-1} Z_{t-1}$$

$$= (Z_{t-1}^\top + \epsilon_t^\top S_{i(t)}) B_{t-1}^{-1} (Z_{t-1} + S_{i(t)}^\top \epsilon_t) - Z_{t-1}^\top B_t^{-1} Z_{t-1}$$

$$\quad - (Z_{t-1}^\top + \epsilon_t^\top S_{i(t)}) B_{t-1}^{-1} S_{i(t)}^\top (I + S_{i(t)} B_{t-1}^{-1} S_{i(t)}^\top)^{-1} S_{i(t)} B_{t-1}^{-1} (Z_{t-1} + S_{i(t)}^\top \epsilon_t)$$

$$= \epsilon_t^\top S_{i(t)} B_{t-1}^{-1} S_{i(t)}^\top \epsilon_t + 2 Z_{t-1}^\top B_{t-1}^{-1} S_{i(t)}^\top \epsilon_t$$

$$\quad - (Z_{t-1}^\top + \epsilon_t^\top S_{i(t)}) B_{t-1}^{-1} S_{i(t)}^\top (I + S_{i(t)} B_{t-1}^{-1} S_{i(t)}^\top)^{-1} S_{i(t)} B_{t-1}^{-1} (Z_{t-1} + S_{i(t)}^\top \epsilon_t)$$

$$= \epsilon_t^\top C_t \epsilon_t + 2 d_t^\top \epsilon_t - (d_t^\top + \epsilon_t^\top C_t)(I + C_t)^{-1}(d_t + C_t \epsilon_t)$$

$$= \epsilon_t^\top C_t (I - (I + C_t)^{-1} C_t) \epsilon_t + 2 d_t^\top (I - (I + C_t)^{-1} C_t) \epsilon_t - d_t^\top (I + C_t)^{-1} d_t$$

$$= \epsilon_t^\top C_t (I + C_t)^{-1} \epsilon_t + 2 d_t^\top (I + C_t)^{-1} \epsilon_t - d_t^\top (I + C_t)^{-1} d_t$$

$$= \left\| \epsilon_t + C_t^{-1} d_t \right\|^2_{C_t(I+C_t)^{-1}} - d_t^\top (I + C_t)^{-1} C_t^{-1} d_t - d_t^\top (I + C_t)^{-1} d_t$$

$$= \left\| \epsilon_t + C_t^{-1} d_t \right\|^2_{C_t(I+C_t)^{-1}} - d_t^\top (I + C_t)^{-1}(I + C_t^{-1}) d_t \,. \tag{56}$$

Therefore, Lemma 11 with $V := \left(C_t(I + C_t)^{-1}\right)^{-1} = (I + C_t)C_t^{-1}$, $v := C_t^{-1} d_t$ yields

$$\mathbb{E}\left[ \exp\left( \frac{1}{2}\left( \|\hat{p}_t - p^*\|^2_{B_t} - \|\hat{p}_{t-1} - p^*\|^2_{B_{t-1}} \right) \right) \right]$$

$$\leq \frac{\sqrt{|(I + C_t)C_t^{-1}|}}{\sqrt{|(I + C_t)C_t^{-1} - I|}} e^{\frac{1}{2} d_t^\top C_t^{-1}((I+C_t)C_t^{-1}-I)^{-1} C_t^{-1} d_t} e^{-\frac{1}{2} d_t^\top (I+C_t)^{-1}(I+C_t^{-1}) d_t}$$

$$\leq \frac{\sqrt{|(I + C_t)C_t^{-1}|}}{\sqrt{|C_t^{-1}|}} e^{\frac{1}{2} d_t^\top C_t^{-1}(C_t^{-1})^{-1} C_t^{-1} d_t} e^{-\frac{1}{2} d_t^\top (I+C_t)^{-1}(I+C_t^{-1}) d_t}$$

$$= \sqrt{|(I + C_t)|}$$

$$= \sqrt{\frac{|B_t|}{|B_{t-1}|}} \,, \tag{57}$$

where see, *e.g.,* Abbasi-Yadkori et al. (2011, Lemma 11) for the last equality. $\qquad\square$

### E.5 Norms under Perturbations

In the following two lemmas, we give some analysis of norms under perturbations.

**Lemma 13.** *Let $A$ be a positive definite matrix. Let $a \in \mathbb{R}^d$ and $\epsilon > 0$ be such that $\epsilon < \|a\|/3$. Then*

$$\min_{x:\|x\| \leq 2\epsilon} \max_{x':\|x'\| \leq \epsilon} \left\{ (a + x + x')^\top A(a + x + x') \right\} = \min_{x'':\|x''\| \leq \epsilon} \left\{ (a + x'')^\top A(a + x'') \right\}. \tag{58}$$

*Proof.* By considering the Lagrangian multiplier we see that any stationary point of the function $(a + x'')^\top A(a + x'')$ over $\{(x, x') : \|x\| \leq 2\epsilon, \|x'\| \leq \epsilon\}$ satisfies

$$A(a + x + x') - \lambda_1 x = 0 \,,$$

$$A(a + x + x') - \lambda_2 x' = 0 \,,$$

$$x^\top x = 4\epsilon^2 \,,$$

$$x'^\top x' = \epsilon^2 \,, \tag{59}$$

and therefore $\lambda_1 x = \lambda_2 x'$. Considering the last two conditions of (59) we have $\lambda_2 = \pm 2\lambda_1$, implying that

$$x' = -(3A - 2\lambda_1 I)Aa \tag{60}$$

or

$$x' = (A - 2\lambda_1 I)Aa \tag{61}$$

for $\lambda_1$ satisfying $x'^\top x' = \epsilon^2$.

Note that it holds for any positive definite matrix $B$ that

$$\frac{\mathrm{d}^2}{\mathrm{d}\lambda^2} a(B + \lambda I)^{-2} a = a(B + \lambda I)^{-4} a = \left\| (B + \lambda I)^{-2} a \right\|^2 , \tag{62}$$

which is positive almost everywhere, meaning that $a(B + \lambda I)^{-2} a$ is strictly convex with respect to $\lambda \in \mathbb{R}$. Therefore, there exists at most two $\lambda_1'$'s satisfying (60) and $x'^\top x' = \epsilon^2$, and there exists at most two $\lambda_1'$'s satisfying (61) and $x'^\top x' = \epsilon^2$. In summary, there at most four stationary points of $(a + x'')^\top A(a + x'')$ over $\{(x, x') : \|x\| \leq 2\epsilon, \|x'\| \leq \epsilon\}$.

On the other hand, two optimization problems

$$\min_{x:\|x\|\leq 2\epsilon} \min_{x':\|x'\|\leq\epsilon} \left\{ (a + x + x')^\top A(a + x + x') \right\} = \min_{x'':\|x''\|\leq 3\epsilon} \left\{ (a + x'')^\top A(a + x'') \right\} \tag{63}$$

and

$$\max_{x:\|x\|\leq 2\epsilon} \max_{x':\|x'\|\leq\epsilon} \left\{ (a + x + x')^\top A(a + x + x') \right\} = \max_{x'':\|x''\|\leq 3\epsilon} \left\{ (a + x'')^\top A(a + x'') \right\} \tag{64}$$

can be easily solved by an elementary calculation and the optimal values are equal to those corresponding to (60).

Therefore, the optimal solutions of the two minimax problems

$$\max_{x:\|x\|\leq 2\epsilon} \min_{x':\|x'\|\leq\epsilon} \left\{ (a + x + x')^\top A(a + x + x') \right\} \tag{65}$$

and

$$\min_{x:\|x\|\leq 2\epsilon} \max_{x':\|x'\|\leq\epsilon} \left\{ (a + x + x')^\top A(a + x + x') \right\} \tag{66}$$

correspond to two points corresponding to (61).

We can see again from an elementary calculation that the optimal solutions for two optimization problems

$$\min_{x'':\|x''\|\leq\epsilon} \left\{ (a + x'')^\top A(a + x'') \right\}$$
$$\max_{x'':\|x''\|\leq\epsilon} \left\{ (a + x'')^\top A(a + x'') \right\} \tag{67}$$

have the same necessary and sufficient conditions as (61) and we complete the proof by noticing that (65) is less than (66). $\qquad\square$

**Lemma 14.** *Let $A \succeq nS_1^\top S_1$ be a positive-definite matrix with minimum eigenvalue at least $\lambda > 0$. Then, for any $\hat{p} \in \mathbb{R}^d$ and $\epsilon > 0$ satisfying $\epsilon < \|\hat{p} - p^*\| / 3$,*

$$\|\hat{p} - p^*\|_A^2 - \inf_{p:\|p-p^*\|\leq 2\epsilon} \sup_{p':\|p'-p\|\leq\epsilon} \|p' - \hat{p}\|_A^2 \geq \epsilon\sqrt{n\lambda} \|S_1(\hat{p} - p^*)\| . \tag{68}$$

*Proof.* Let $a = \hat{p} - p^*$. By Lemma 13, we have

$$\inf_{p:\|p-p^*\|\leq 2\epsilon} \sup_{p':\|p'-p\|\leq\epsilon} \|p' - \hat{p}\|_A^2$$
$$= \inf_{x:\|x\|\leq 2\epsilon} \sup_{x':\|x'\|\leq\epsilon} \|a + x + x'\|_A^2$$
$$= \inf_{x:\|x\|\leq\epsilon} \|a + x\|_A^2 . \tag{69}$$

Now define $\mathcal{S}_{\epsilon',A} = \{x : \|x\|_A \leq \epsilon'\}$. Then, we see that $\mathcal{S}_{\epsilon\sqrt{\lambda},A} \subset \{x : \|x\| \leq \epsilon\}$. Therefore, an elementary calculation using the Lagrange multiplier technique shows

$$\inf_{x:\|x\|\leq\epsilon} \|p' - \hat{p}\|_A^2 \leq \inf_{x\in\mathcal{S}_{\epsilon\sqrt{\lambda},A}} \|p - \hat{p}\|_A^2$$
$$= \left(\|a\|_A - \epsilon\sqrt{\lambda}\right)^2 . \tag{70}$$

As a result, we see that

$$\|p^* - \hat{p}\|_A^2 - \inf_{p:\|p-p^*\|\leq 2\epsilon} \sup_{p':\|p'-p\|\leq\epsilon} \|p' - \hat{p}\|_A^2 \geq \|a\|_A^2 - \left(\|a\|_A - \epsilon\sqrt{\lambda}\right)^2$$
$$= \epsilon\sqrt{\lambda}\left(\|a\|_A + \|a\|_A - \epsilon\sqrt{\lambda}\right)$$
$$\geq \epsilon\sqrt{\lambda}\left(\|a\|_A + \|a\|\sqrt{\lambda} - \epsilon\sqrt{\lambda}\right)$$
$$= \epsilon\sqrt{\lambda}\left(\|a\|_A + \sqrt{\lambda}(\|a\| - \epsilon)\right)$$
$$\geq \epsilon\sqrt{\lambda}\|a\|_A$$
$$\geq \epsilon\sqrt{n\lambda}\|S_1 a\| . \tag{71}$$

$\square$

For the subsets of $\mathbb{R}^n$, $\mathcal{X}$ and $\mathcal{Y}$, let $\mathcal{X} + \mathcal{Y} := \{x + y : x \in \mathcal{X}, y \in \mathcal{Y}\}$ be the Minkowski sum, and let $B_r^n(p)$ be the $n$-dimensional Euclidian ball of radius $r$ at point $p \in \mathbb{R}^n$ (the superscript $n$ can be omitted when it is clear from context). We also let $\epsilon'$ be

$$\epsilon' := \frac{\epsilon}{\left(16 \max_{i\in[N]}\|S_i\|\right) \vee \left(\frac{1}{\sqrt{A}} \max_{i\in[N]} \frac{\|L_i - L_1\|}{\|z_{1,i}\|}\right)} , \tag{72}$$

which is also used throughout the proof of this appendix and Appendix F as $\epsilon$ in (26).

**Theorem 15.** *Let $\epsilon'' \in (0, \epsilon)$ be a constant for $\epsilon$ defined in (26). Let $\hat{p} \in \mathcal{C}_k + B_{\epsilon'}^d(0)$ be satisfying $\|S_k(\hat{p} - p^*)\| \leq \epsilon''$. Then, there exists $\delta > 0$ satisfying for any $n \geq 0$ and $A \succeq nS_1^\top S_1 + \lambda I$ that*

$$\|p^* - \hat{p}\|_A^2 - \inf_{p:\|p-p^*\|\leq 2\epsilon} \sup_{p':\|p'-p\|\leq\epsilon} \|p' - \hat{p}\|_A^2 \geq \epsilon\delta\sqrt{\lambda n} . \tag{73}$$

*Proof.* Recall that $\epsilon'' < \epsilon \leq \min_{p\in\mathcal{C}_1^c} \|p - p^*\| / 3$. It is enough from Lemma 14 to prove that

$$\delta := \min_{\hat{p}\in\{p\in\mathcal{C}_k+B_{\epsilon'}^d(0):\|S_k(p-p^*)\|\leq\epsilon''\}} \|S_1(\hat{p} - p^*)\| \tag{74}$$

is positive.

We prove by contradiction and the proof is basically same as that of Lemma 7 but more general in the sense that the condition on $\hat{p}$ is not $\hat{p} \in \mathcal{C}_k$ but $\hat{p} \in \mathcal{C}_k + B_{\epsilon'}^d(0)$. Assume that $\delta = 0$, that is, there exists $\hat{p} \in \mathcal{C}_k + B_{\epsilon'}^d(0)$ satisfying $\|S_k(p-p^*)\| \leq \epsilon''\}$ and $\|S_1(\hat{p} - p^*)\| = 0$. Note that $\|S_1(\hat{p} - p^*)\| = 0$ implies $\|S_1(\hat{p} - p^*)\| \leq \epsilon''$. Therefore, we now have following conditions on $\hat{p}$:

- $\hat{p} \in \mathcal{C}_k + B_{\epsilon'}^d(0)$
- $\|S_1(\hat{p} - p^*)\| \leq \epsilon''$
- $\|S_k(\hat{p} - p^*)\| \leq \epsilon''$ .

Following the same argument as the proof of Lemma 7, we have

$$z_{1,k}^\top \begin{pmatrix} S_1 \\ S_k \end{pmatrix} (\hat{p} - p^*) \leq \sqrt{2A}\epsilon''\|z_{1,k}\| . \tag{75}$$

On the other hand, since $\hat{p} \in \mathcal{C}_k + B_{\epsilon'}^d(0)$ we can take $\bar{p} \in \mathcal{C}_k$ such that $\|\hat{p} - \bar{p}\| \leq \epsilon'$. Hence,

$$
\begin{aligned}
z_{1,k}^\top \begin{pmatrix} S_1 \\ S_k \end{pmatrix} (\hat{p} - p^*) &= (L_1 - L_k)^\top (\hat{p} - p^*) \\
&= -(L_k - L_1)^\top (\hat{p} - \bar{p}) + (L_1 - L_k)^\top \bar{p} + (L_k - L_1)^\top p^* \\
&\geq -(L_k - L_1)^\top (\hat{p} - p^*) + \Delta_k . \quad \text{(by } \bar{p} \in \mathcal{C}_k \text{ and def. of } \Delta_k)
\end{aligned}
$$
(76)

From (75) and (76), we have

$$
\Delta_k - (L_k - L_1)^\top (\hat{p} - p^*) \leq \sqrt{2A}\epsilon'' \|z_{1,k}\| .
$$
(77)

Now, the left hand side of (77) is bounded from below as

$$
\begin{aligned}
\Delta_k - (L_k - L_1)^\top (\hat{p} - \bar{p}) &\geq \Delta_k - \|L_k - L_1\|\|\hat{p} - \bar{p}\| \\
&\geq \Delta_k - \|L_k - L_1\|\epsilon' \\
&= \Delta_k - \|L_k - L_1\| \frac{\epsilon}{\frac{1}{\sqrt{A}} \max_i \frac{\|L_1 - L_i\|}{\|z_{1,i}\|}} \\
&= \Delta_k - \|L_k - L_1\| \frac{\frac{1}{2\sqrt{A}} \min_i \frac{\Delta_i}{\|z_{1,i}\|}}{\frac{1}{\sqrt{A}} \max_i \frac{\|L_1 - L_i\|}{\|z_{1,i}\|}} \\
&\geq \Delta_k - \Delta_k/2 .
\end{aligned}
$$
(78)

On the other hand, using the definition of $\epsilon''$, the right hand side of (77) is bounded from above as

$$
\sqrt{2A}\epsilon'' \|z_{1,k}\| < \Delta_k/2 .
$$
(79)

Therefore, the proof is completed by contradiction. $\qquad\square$

### E.6 Exit Time Analysis

We next consider the exit time. Let $\mathcal{A}_t$ be an event deterministic given $\mathcal{F}_t$, and $\mathcal{B}_t$ be a random event such that if $\mathcal{B}_t$ occurred then $\mathcal{A}_{t'}$ never occurs for $t' = t + 1, t + 2, \dots$. Let $P_t$, $t = 1, 2, \dots, T$, be a stochastic process satisfying $P_t \leq \mathbb{P}\{\mathcal{B}_t|\mathcal{F}_t\}$ a.s. and $P_t^{-1}$ is a supermartingale with respect to the filtration induced by $\mathcal{F}_t$.

**Theorem 16.** *Let $\tau$ be the stopping time defined as*

$$
\tau = \begin{cases} \min\{t \in [T] : \mathcal{A}_t\} & \text{if } \mathcal{A}_t \text{ occurs for some } t \in [T]. \\ T + 1 & \text{otherwise.} \end{cases}
$$
(80)

*Then we almost surely have*

$$
\mathbb{E}\left[ \sum_{t=1}^T \mathbb{1}[\mathcal{A}_t] \,\middle|\, \mathcal{F}_\tau \right] \leq \begin{cases} P_\tau^{-1} & \tau \leq T, \\ 0 & \tau = T + 1. \end{cases}
$$
(81)

We prove this theorem based on the following lemma.

**Lemma 17.** *Let $(Q_i)_{i=1}^\infty \subset [0, 1]$ be an arbitrary stochastic process such that $(Q_i^{-1})_{i=1}^\infty$ is a supermartingale with respect to a filtration $(\mathcal{G}_i)_{i=1}^\infty$. Then, for any $\mathcal{G}_0 \subset \mathcal{G}_1$,*

$$
\mathbb{E}\left[ \sum_{i=1}^T \prod_{j=1}^i (1 - Q_j) \,\middle|\, \mathcal{G}_0 \right] \leq \mathbb{E}\left[ Q_1^{-1}|\mathcal{G}_0 \right] - 1 \quad \text{a.s.}
$$
(82)

*Proof.* Let

$$
N_k((Q_i, \mathcal{G}_i)_{i=1}^\infty, \mathcal{G}_0) = \mathbb{E}\left[ \sum_{i=1}^k \prod_{j=1}^i (1 - Q_j) \,\middle|\, \mathcal{G}_0 \right]
$$

$$
\overline{N}_k((Q_i, \mathcal{G}_i)_{i=1}^\infty, \mathcal{G}_0) = \mathbb{E}\left[ \sum_{i=1}^\infty \prod_{j=1}^i (1 - Q_j) \,\middle|\, \mathcal{G}_0 \right] \quad \text{where } Q_j = Q_k \text{ for } j > k.
$$
(83)

We show $\overline{N}_k((Q_i, \mathcal{G}_i)_{i=1}^\infty, \mathcal{G}_0) \leq \mathbb{E}[Q_1^{-1}|\mathcal{G}_0] - 1$ a.s. for any $(Q_i, \mathcal{G}_i)_{i=1}^\infty$, $\mathcal{G}_0 \subset \mathcal{G}_1$ and $k \in \mathbb{N}$ by induction. First, for $k = 1$ the statement holds since

$$\overline{N}_1((Q_i, \mathcal{G}_i)_{i=1}^\infty, \mathcal{G}_0) = \mathbb{E}\left[\sum_{i=1}^\infty \prod_{j=1}^i (1 - Q_1) \,\middle|\, \mathcal{G}_0\right]$$
$$= \mathbb{E}\left[Q_1^{-1} - 1 \,\middle|\, \mathcal{G}_0\right]$$
$$= \mathbb{E}\left[Q_1^{-1} \,\middle|\, \mathcal{G}_0\right] - 1 \tag{84}$$

Next, assume that the statement holds for all $(Q_i, \mathcal{G}_i)_{i=1}^k$, $\mathcal{G}_0 \subset \mathcal{G}_1$ and $k \leq k_0$. Then, we almost surely have

$$\overline{N}_{k_0+1}((Q_i, \mathcal{G}_i)_{i=1}^\infty, \mathcal{G}_0) = \mathbb{E}\left[(1 - Q_1)\mathbb{E}\left[1 + \sum_{i=2}^\infty \prod_{j=2}^i (1 - Q_j) \,\middle|\, \mathcal{G}_1\right] \,\middle|\, \mathcal{G}_0\right]$$
$$= \mathbb{E}\left[(1 - Q_1)(1 + \overline{N}_{k_0}((Q_i, \mathcal{G}_i)_{i=2}^\infty, \mathcal{G}_1)) \,\middle|\, \mathcal{G}_0\right]$$
$$\leq \mathbb{E}\left[(1 - Q_1)\mathbb{E}[Q_2^{-1} \,\middle|\, \mathcal{G}_1] \,\middle|\, \mathcal{G}_0\right] \quad \text{(assumption of the induction)}$$
$$\leq \mathbb{E}\left[Q_1^{-1} \,\middle|\, \mathcal{G}_0\right] - 1 \quad (Q_i^{-1} \text{ is a supermartingale.}) \tag{85}$$

We obtain the lemma from

$$\mathbb{E}\left[\sum_{i=1}^k \prod_{j=1}^i (1 - Q_j) \,\middle|\, \mathcal{G}_0\right] = N_k((Q_i, \mathcal{G}_i)_{i=1}^\infty, \mathcal{G}_0) \leq \overline{N}_k((Q_i, \mathcal{G}_i)_{i=1}^\infty, \mathcal{G}_0) \quad \text{a.s.} \tag{86}$$

$\square$

*Proof of Theorem 16.* The statement is obvious for the case $\tau = T + 1$ and we consider the other case in the following.

Let $\tau_i$ be the time of the $i$-th occurrence of $\mathcal{A}_t$. More formally, we define $\tau_i$ as the stopping time $\tau_1 = \tau$ and

$$\tau_{i+1} = \begin{cases} \min\left\{t \in [T] : \sum_{t'=1}^T \mathbb{1}[\mathcal{A}_{t'}] = i + 1\right\} & \sum_{t'=1}^T \mathbb{1}[\mathcal{A}_{t'}] \geq i + 1, \\ \tau_i + 1 & \text{otherwise.} \end{cases} \tag{87}$$

Then $(P_i') = (P_{\tau_i})$ is a stochastic process measurable by the filtration induced by $(\mathcal{F}_i') = (\mathcal{F}_{\tau_i})$. By Lemma 17 we obtain

$$\mathbb{E}\left[\sum_{t=1}^T \mathbb{1}[\mathcal{A}_t] \,\middle|\, \mathcal{F}_\tau\right] = \mathbb{E}\left[\sum_{n=1}^T \mathbb{1}\left[\sum_{t=1}^T \mathbb{1}[\mathcal{A}_t] \geq n \,\middle|\, \mathcal{F}_\tau\right]\right]$$
$$\leq 1 + \mathbb{E}\left[\sum_{n=2}^T \mathbb{1}\left[\sum_{t=1}^T \mathbb{1}[\mathcal{A}_t] \geq n \,\middle|\, \mathcal{F}_\tau\right]\right]$$
$$\leq 1 + \mathbb{E}\left[\sum_{i=1}^T \prod_{j=1}^i (1 - P_j') \,\middle|\, \mathcal{F}_1'\right]$$
$$\leq 1 + \mathbb{E}\left[(P_1')^{-1}|\mathcal{F}_1'\right] - 1$$
$$= P_\tau^{-1}. \tag{88}$$

$\square$

# F Regret Analysis of TSPM Algorithm

In this appendix, we give the proof of Theorem 3. Note that the cells are defined for the decomposition of $\mathbb{R}^M$, not $\mathcal{P}_M$. In other words, the cell $\mathcal{C}_i$ is here defined as $\mathcal{C}_i = \{p \in \mathbb{R}^M : \text{action } i \text{ is optimal}\}$.

For the linear setting, the empirical feedback distribution $q_i^{(t)}$ and $q_{i,n}$ are defined as

$$q_i^{(t)} := \frac{1}{N_i(t)} \sum_{s \in [t-1]:i(s)=i} y(s), \tag{89}$$

$$q_{i,n} := \text{the value of } q_i^{(t)} \text{ after taking action } i \text{ for } n \text{ times.} \tag{90}$$

Recall that $\hat{p}_t = B_t^{-1} b_t$, which is the mode of $\bar{g}_t(p)$.

## F.1 Regret Decomposition

Here, we break the regret into several terms. For any $i \in [N]$, we define events

$$\mathcal{A}_i(t) := \left\{ \|S_i \hat{p}_t - S_i p^*\| \leq \frac{\epsilon}{4} \right\}, \tag{91}$$

$$\tilde{\mathcal{A}}_i(t) := \{ \|S_i \tilde{p}_t - S_i p^*\| \leq \epsilon \}. \tag{92}$$

We first decompose the regret as

$$\begin{aligned}
\text{Reg}(T) &= \sum_{t=1}^{T} \Delta_{i(t)} \\
&\leq \sum_{t=1}^{T} \left( \Delta_{i(t)} \mathbb{1}\left[\tilde{\mathcal{A}}_1(t)\right] + \max_{j \in [N]} \Delta_j \mathbb{1}\left[\tilde{\mathcal{A}}_1^c(t)\right] \right) \\
&= \sum_{i \neq 1} \sum_{t=1}^{T} \Delta_i \mathbb{1}\left[i(t) = i, \tilde{\mathcal{A}}_1(t)\right] + \max_{j \in [N]} \Delta_j \sum_{t=1}^{T} \mathbb{1}\left[\tilde{\mathcal{A}}_1^c(t)\right] \\
&\leq \sum_{i \neq 1} \Delta_i \sum_{t=1}^{T} \left( \underbrace{\mathbb{1}\left[i(t) = i, \tilde{\mathcal{A}}_1(t), \mathcal{A}_i(t)\right]}_{(A)} + \underbrace{\mathbb{1}[i(t) = i, \mathcal{A}_i^c(t)]}_{(B)} \right) + \max_{j \in [N]} \Delta_j \sum_{t=1}^{T} \mathbb{1}\left[\tilde{\mathcal{A}}_1^c(t)\right].
\end{aligned} \tag{93}$$

To decompose the last term, we define the following notation. We define for any $i \in [N]$

$$P_i(t) := \mathbb{P}\{\tilde{p}_t \in \mathcal{C}_i \mid \mathcal{F}_t\}. \tag{94}$$

We also define

$$\mathcal{C}_{i,t} := \mathcal{C}_i \cap B_{\epsilon'}(\hat{p}_t), \tag{95}$$

where $\epsilon'$ is defined in (72), and

$$\bar{i}_t := \arg\max_{i \in [N]} \mathbb{P}\{\tilde{p}_t \in \mathcal{C}_{i,t} \mid \mathcal{F}_t\}. \tag{96}$$

We define $\bar{p}_t$ as an arbitrary point in $\mathcal{C}_{\bar{i}_t,t}$. Then, we define

$$\bar{\mathcal{A}}_i(t) := \left\{ \|S_i \bar{p}_t - S_i p^*\| \leq \frac{\epsilon}{8} \right\}. \tag{97}$$

Using these notations, the last term in (93) can be decomposed as

$$\begin{aligned}
\mathbb{1}\left[\tilde{\mathcal{A}}_1^c(t)\right] &\leq \sum_{k=1}^{N} \mathbb{1}\left[\bar{p}_t \in \mathcal{C}_k, \tilde{\mathcal{A}}_1^c(t)\right] \\
&= \sum_{k=1}^{N} \mathbb{1}\left[\bar{p}_t \in \mathcal{C}_k, \bar{\mathcal{A}}_k^c(t), \tilde{\mathcal{A}}_1^c(t)\right] + \sum_{k=1}^{N} \mathbb{1}\left[\bar{p}_t \in \mathcal{C}_k, \bar{\mathcal{A}}_k(t), \tilde{\mathcal{A}}_1^c(t)\right] \\
&\leq \underbrace{\sum_{k=1}^{N} \mathbb{1}\left[\bar{p}_t \in \mathcal{C}_k, \bar{\mathcal{A}}_k^c(t)\right]}_{(C)} + \underbrace{\mathbb{1}\left[\bar{p}_t \in \mathcal{C}_1, \bar{\mathcal{A}}_1(t), \tilde{\mathcal{A}}_1^c(t)\right]}_{(D)} + \underbrace{\sum_{k=2}^{N} \mathbb{1}\left[\bar{p}_t \in \mathcal{C}_k, \bar{\mathcal{A}}_k(t)\right]}_{(E)}.
\end{aligned} \tag{98}$$

We will bound the expectation of each term in the following and complete the proof of Theorem 3 as

$$
\begin{aligned}
\mathbb{E}\left[\mathrm{Reg}(T)\right] &= \sum_{i \neq 1} \Delta_i \left( \mathrm{O}\left(\frac{1}{\epsilon^2}\log T\right) + \mathrm{O}\left(\frac{N}{\epsilon^2}\log T\right) \right) \\
&\quad + \max_{j \in [N]} \Delta_j \left( \sum_{k=1}^{N} \mathrm{O}\left(\frac{NM}{\epsilon^2}\log T\right) + \mathrm{O}(1) + \sum_{k=2}^{N} \mathrm{O}(1) \right) \\
&= \mathrm{O}\left( \max\left\{ \frac{N\sum_{i\in[N]}\Delta_i}{\epsilon^2}, \frac{N^2 M \max_{i\in[N]}\Delta_i}{\epsilon^2} \right\} \log T \right) \\
&= \mathrm{O}\left( \frac{A N^2 M \max_{i\in[N]}\Delta_i}{\Lambda^2} \log T \right),
\end{aligned}
\tag{99}
$$

where the last transformation follows from the definition of $\epsilon$ in (26).

## F.2 Analysis for Case (A)

**Lemma 18.** *For any $i \neq 1$,*

$$
\mathbb{E}\left[ \sum_{t=1}^{T} \mathbb{1}\left[i(t) = i,\, \tilde{\mathcal{A}}_1(t),\, \mathcal{A}_i(t)\right] \right] \leq \frac{64}{9\epsilon^2}\log T + 2^{M/2}.
\tag{100}
$$

To prove Lemma 18, we prove the following lemma using Corollary 8 and Lemma 9.

**Lemma 19.** *For any $0 \leq \xi < 1/2$,*

$$
\mathbb{P}\{\tilde{p}_t \in \mathcal{V}_i \mid \mathcal{A}_i(t),\, N_i(t) > n_i\} \leq \exp\left(-\frac{9}{16}\xi\epsilon^2 n_i\right)(1 - 2\xi)^{-M/2},
\tag{101}
$$

*where $\mathcal{V}_i := \{p \in \mathcal{C}_i : \|S_1 p - S_1 p^*\| \leq \epsilon\}$.*

*Proof.* Since $\tilde{p}_t \sim \mathcal{N}(\hat{p}_t, B_t^{-1})$ for $\hat{p}_t = B_t^{-1} b_t$, the squared Mahalanobis distance $\|B_t^{1/2}(\tilde{p}_t - \hat{p}_t)\|^2$ follows the chi-squared distribution with $M$ degree of freedom. Therefore, we have

$$
\mathbb{P}\{\tilde{p}_t \in \mathcal{V}_i \mid \mathcal{A}_i(t),\, N_i(t) > n_i\} \leq h\left( \inf_{p \in \mathcal{V}_i} \|B_t^{1/2}(p - \hat{p}_t)\|^2 \right),
\tag{102}
$$

where $h(a) = \mathbb{P}_{X \sim \chi_M^2}\{X \geq a\}$. To use Lemma 9, we check the condition of Lemma 9 is indeed satisfied. First, it is obvious that the assumptions $N_i(t) \geq n_i$ and $\|S_i \hat{p}_t - S_i p^*\| < \epsilon/4$ are satisfied. Besides, $p \in \mathcal{V}_i$ implies $p \in \mathcal{T}_i = \{p \in \mathbb{R}^M : \|S_i p - S_i p^*\| \geq \epsilon\}$ from Corollary 8. Thus, applying Lemma 9 concludes the proof. □

*Proof of Lemma 18.* For any $n_i > 0$,

$$
\begin{aligned}
&\sum_{t=1}^{T} \mathbb{1}\left[i(t) = i,\, \tilde{\mathcal{A}}_1(t),\, \mathcal{A}_i(t)\right] \\
&= \sum_{t=1}^{T} \mathbb{1}\left[i(t) = i,\, \tilde{\mathcal{A}}_1(t),\, \mathcal{A}_i(t),\, N_i(t) \leq n_i\right] + \sum_{t=1}^{T} \mathbb{1}\left[i(t) = i,\, \tilde{\mathcal{A}}_1(t),\, \mathcal{A}_i(t),\, N_i(t) > n_i\right] \\
&\leq n_i + \sum_{t=1}^{T} \mathbb{1}\left[i(t) = i,\, \tilde{\mathcal{A}}_1(t),\, \mathcal{A}_i(t),\, N_i(t) > n_i\right].
\end{aligned}
\tag{103}
$$

The second term is bounded from above as

$$
\mathbb{E}\left[\sum_{t=1}^{T} \mathbb{1}\Big[i(t) = i,\, \tilde{\mathcal{A}}_1(t),\, \mathcal{A}_i(t),\, N_i(t) > n_i\Big]\right]
$$

$$
= \sum_{t=1}^{T} \mathbb{P}\Big\{i(t) = i,\, \tilde{\mathcal{A}}_1(t),\, \mathcal{A}_i(t),\, N_i(t) > n_i\Big\}
$$

$$
\leq \sum_{t=1}^{T} \mathbb{P}\Big\{i(t) = i,\, \tilde{\mathcal{A}}_1(t) \,\Big|\, \mathcal{A}_i(t),\, N_i(t) > n_i\Big\}
$$

$$
= \sum_{t=1}^{T} \mathbb{P}\Big\{i(t) = i,\, \tilde{\mathcal{A}}_1(t),\, \tilde{p}_t \in \mathcal{C}_i \,\Big|\, \mathcal{A}_i(t),\, N_i(t) > n_i\Big\} \quad (i(t) = i \text{ implies } \tilde{p}_t \in \mathcal{C}_i)
$$

$$
\leq \sum_{t=1}^{T} \mathbb{P}\{\tilde{p}_t \in \mathcal{V}_i \mid \mathcal{A}_i(t),\, N_i(t) > n_i\}. \tag{104}
$$

To obtain an upper bound for $\mathbb{P}\{\tilde{p}_t \in \mathcal{V}_i \mid \mathcal{A}_i(t),\, N_i(t) > n_i\}$, we use Lemma 19. By taking $n_i = \frac{16}{9}\frac{1}{\xi\epsilon^2}\log T$ with $\xi = 1/4$, we have

$$
\mathbb{E}\left[\sum_{t=1}^{T} \mathbb{1}\Big[i(t) = i,\, \tilde{\mathcal{A}}_1(t),\, \mathcal{A}_i(t)\Big]\right] \leq n_i + \sum_{t=1}^{T} \mathbb{P}\{\tilde{p}_t \in \mathcal{V}_i \mid \mathcal{A}_i(t),\, N_i(t) > n_i\}
$$

$$
\leq n_i + \sum_{t=1}^{T} \exp\Big(-\frac{9}{16}\xi\epsilon^2 n_i\Big)(1 - 2\xi)^{-M/2} \quad \text{(by Lemma 19)}
$$

$$
= \frac{16}{9}\frac{1}{\xi\epsilon^2}\log T + (1 - 2\xi)^{-M/2}
$$

$$
= \frac{64}{9\epsilon^2}\log T + 2^{M/2}, \tag{105}
$$

which completes the proof. $\qquad\square$

## F.3 Analysis for Case (B)

**Lemma 20.** *For any $i \neq 1$,*

$$
\mathbb{E}\left[\sum_{t=1}^{T} \mathbb{1}[i(t) = i,\, \mathcal{A}_i^c(t)]\right] \leq \frac{256N\left(\log T + \frac{A}{2}\log 2 + 1\right)}{\epsilon^2} + \frac{16A^2}{\epsilon^2} \tag{106}
$$

The regret in this case can intuitively be bounded because as the round proceeds the event $i(t) = i$ makes $S_i\hat{p}_t$ close to $S_i p^*$, which implies that the expected number of times the event $\mathcal{A}_i^c(t)$ occurs is not large.

Before going to the analysis of Lemma 20, we prove useful inequalities between $\|q_i^{(t)} - S_i p^*\|$, $\|q_i^{(t)} - S_i\hat{p}_t\|$, and $\|S_i\hat{p}_t - S_i p^*\|$.

**Lemma 21.** *Assume $N_i(t) > 0$. Then,*

$$
\|q_i^{(t)} - S_i\hat{p}_t\|^2 \leq \frac{Z_{\setminus i}}{N_i(t)} + \|q_i^{(t)} - S_i p^*\|^2. \tag{107}
$$

*Proof.* Recall that $\hat{p}_t$ is the maximizer of $\bar{g}_t(p)$, and we have

$$
\hat{p}_t = \arg\max_{p \in \mathbb{R}^M} \bar{g}_t(p) = \arg\max_{p \in \mathbb{R}^M} \prod_{i=1}^{N} \exp\Big\{-\frac{1}{2}N_i(t)\|q_i^{(t)} - S_i p\|^2\Big\} = \arg\min_{p \in \mathbb{R}^M} \sum_{i=1}^{N} N_i(t)\|q_i^{(t)} - S_i p\|^2.
$$

$$
\tag{108}
$$

Using this and the definition of $Z_{\setminus i}$, we have

$$
\begin{aligned}
N_i(t)\|q_i^{(t)} - S_i\hat{p}_t\|^2 &\leq \sum_{k \in [N]} N_k(t)\|q_k^{(t)} - S_k\hat{p}_t\|^2 \\
&\leq \sum_{k \in [N]} N_k(t)\|q_k^{(t)} - S_k p^*\|^2 \\
&\leq Z_{\setminus i} + N_i(t)\|q_i^{(t)} - S_i p^*\|^2 .
\end{aligned}
\tag{109}
$$

Dividing by $N_i(t)$ on the both sides completes the proof. $\qquad\square$

**Lemma 22.** *Assume that $\mathcal{A}_i^c(t)$ and $N_i(t) > 0$ hold. Then,*

$$
\|q_i^{(t)} - S_i p^*\| > \frac{1}{2}\left(\frac{\epsilon}{4} - \sqrt{\frac{Z_{\setminus i}}{N_i(t)}}\right).
\tag{110}
$$

*Proof.* By the triangle inequality,

$$
\begin{aligned}
\|q_i^{(t)} - S_i p^*\| &\geq \|S_i\hat{p}_t - S_i p^*\| - \|q_i^{(t)} - S_i\hat{p}_t\| \\
&> \frac{\epsilon}{4} - \sqrt{\frac{Z_{\setminus i}}{N_i(t)} + \|q_i^{(t)} - S_i p^*\|^2} \quad \text{(by $\mathcal{A}_i^c(t)$ and Lemma 21)} \\
&\geq \frac{\epsilon}{4} - \sqrt{\frac{Z_{\setminus i}}{N_i(t)}} - \|q_i^{(t)} - S_i p^*\| \quad \text{(by $\sqrt{x+y} \leq \sqrt{x} + \sqrt{y}$ for $x, y \geq 0$)} ,
\end{aligned}
\tag{111}
$$

which is equivalent to (110). $\qquad\square$

*Proof of Lemma 20.* We first bound the expectation conditioned on $Z_{\setminus i}$, and then take the expectation for $Z_{\setminus i}$. Now,

$$
\begin{aligned}
&\mathbb{E}\left[\sum_{t=1}^{T} \mathbb{1}[i(t) = i, \mathcal{A}_i^c(t)] \,\middle|\, Z_{\setminus i}\right] \\
&= \mathbb{E}\left[\sum_{t=1}^{T} \mathbb{1}\left[i(t) = i, \mathcal{A}_i^c(t), N_i(t) \leq \frac{64 Z_{\setminus i}}{\epsilon^2}\right] \,\middle|\, Z_{\setminus i}\right] \\
&\quad + \mathbb{E}\left[\sum_{t=1}^{T} \mathbb{1}\left[i(t) = i, \mathcal{A}_i^c(t), N_i(t) > \frac{64 Z_{\setminus i}}{\epsilon^2}\right] \,\middle|\, Z_{\setminus i}\right] \\
&\leq \frac{64 Z_{\setminus i}}{\epsilon^2} + \mathbb{E}\left[\sum_{t=1}^{T} \mathbb{1}\left[i(t) = i, \mathcal{A}_i^c(t), N_i(t) > \frac{64 Z_{\setminus i}}{\epsilon^2}\right] \,\middle|\, Z_{\setminus i}\right] \quad (i(t) = i \text{ for all } t \in [T]) .
\end{aligned}
\tag{112}
$$

The first term becomes $256N\left(\log T + \frac{A}{2}\log 2 + 1\right)/\epsilon^2$ by taking expectation over $Z_{\setminus i}$ using Lemma 10. Then, we bound the second term. From Lemma 22, $\mathcal{A}_i^c(t)$ and $N_i(t) > \frac{64 Z_{\setminus i}}{\epsilon^2}$ im-

ply $\|q_i^{(t)} - S_i p^*\| > \epsilon/16$. Therefore,

$$\mathbb{E}\left[\sum_{t=1}^{T} \mathbb{1}\left[i(t) = i,\, \mathcal{A}_i^c(t),\, N_i(t) > \frac{64 Z_{\setminus i}}{\epsilon^2}\right]\,\middle|\, Z_{\setminus i}\right]$$

$$\leq \mathbb{E}\left[\sum_{t=1}^{T} \mathbb{1}\left[i(t) = i,\, \|q_i^{(t)} - S_i p^*\| > \frac{\epsilon}{16}\right]\right]$$

$$\leq \mathbb{E}\left[\sum_{t=1}^{T} \mathbb{1}\left[i(t) = i,\, \bigcup_{y \in [A]} |(q_i^{(t)})_y - (S_i)_y p^*| > \frac{\epsilon}{16\sqrt{A}}\right]\right]$$

$$\leq \mathbb{E}\left[\sum_{y=1}^{A}\sum_{t=1}^{T} \mathbb{1}\left[i(t) = i,\, |(q_i^{(t)})_y - (S_i)_y p^*| > \frac{\epsilon}{16\sqrt{A}}\right]\right]$$

$$\leq \mathbb{E}\left[\sum_{y=1}^{A}\sum_{n_i=1}^{T}\sum_{t=1}^{T} \mathbb{1}\left[i(t) = i,\, N_i(t) = n_i,\, |(q_i^{(t)})_y - (S_i)_y p^*| > \frac{\epsilon}{16\sqrt{A}}\right]\right]$$

$$= \mathbb{E}\left[\sum_{y=1}^{A}\sum_{n_i=1}^{T} \mathbb{1}\left[\bigcup_{t=1}^{T}\left\{i(t) = i,\, N_i(t) = n_i,\, |(q_i^{(t)})_y - (S_i)_y p^*| > \frac{\epsilon}{16\sqrt{A}}\right\}\right]\right]$$

(The event $\{i(t) = i,\, N_i(t) = n_i\}$ occurs at most once for fixed $n_i$.)

$$\leq \sum_{y=1}^{A}\sum_{n_i=1}^{T} \mathbb{P}\left\{|(q_{i,n_i})_y - (S_i)_y p^*| > \frac{\epsilon}{4\sqrt{A}}\right\}$$

$$\leq \sum_{y=1}^{A}\sum_{n_i=1}^{T} 2\exp\left(-2n_i\left(\frac{\epsilon}{4\sqrt{A}}\right)^2\right) \quad \text{(by Hoeffding's ineq.)}$$

$$\leq 2A \sum_{n_i=1}^{\infty} \exp\left(-\frac{n_i \epsilon^2}{8A}\right)$$

$$= 2A \frac{1}{\exp\left(\frac{\epsilon^2}{8A}\right) - 1}$$

$$\leq 2A \frac{1}{\frac{\epsilon^2}{8A}} \quad \text{(by } e^x \geq 1 + x\text{)}$$

$$= \frac{16 A^2}{\epsilon^2}. \tag{113}$$

By summing up the above argument, the proof is completed. $\qquad\square$

### F.4 Analysis for Case (C)

Before going to the analysis of cases (C), (D), and (E), we recall some notations. Recall that

$$P_i(t) = \mathbb{P}\{\tilde{p}_t \in \mathcal{C}_i \mid \mathcal{F}_t\}, \tag{114}$$

$\mathcal{C}_{i,t} = \mathcal{C}_i \cap B_{\epsilon'}(\hat{p}_t)$, $\bar{i}_t = \arg\max_{i \in [N]} \mathbb{P}\{\tilde{p}_t \in \mathcal{C}_{i,t} | \mathcal{F}_t\}$, and $\bar{p}_t$ is an arbitrary point in $\mathcal{C}_{\bar{i}_t, t}$. Also recall that

$$\bar{\mathcal{A}}_i(t) = \left\{\|S_i \bar{p}_t - S_i p^*\| \leq \frac{\epsilon}{8}\right\}. \tag{115}$$

**Lemma 23.** *For any* $i \in [N]$,

$$\mathbb{E}\left[\sum_{t=1}^{T} \mathbb{1}\left[\bar{p}_t \in \mathcal{C}_i,\, \bar{\mathcal{A}}_i^c(t)\right]\right] \leq \frac{N}{p_0}\left(\frac{2^5 M \log T}{\epsilon^2} + e^{\lambda\|p^*\|^2/2}\left(\frac{1}{\lambda T} + \frac{L}{M\lambda}\right)^{M/2}\frac{1}{1 - e^{-\epsilon^2/2^5}}\right). \tag{116}$$

Before proving the above lemma, we give two lemmas.

**Lemma 24.**
$$\mathbb{P}\{\tilde{p}_t \in \mathcal{C}_{\bar{i}_t} \mid \mathcal{F}_t\} \geq \mathbb{P}\{\tilde{p}_t \in \mathcal{C}_{\bar{i}_t,t} \mid \mathcal{F}_t\} \geq p_0/N \,, \tag{117}$$
*where* $p_0 := 1 - h((\lambda\epsilon')^2)$.

*Proof.* First, we prove
$$\mathbb{P}\left\{\tilde{p}_t \in \bigcup_{i \in [N]} \mathcal{C}_{i,t} \;\middle|\; \mathcal{F}_t\right\} \geq 1 - h((\lambda\epsilon')^2) \,. \tag{118}$$

This follows from
$$\mathbb{P}\left\{\tilde{p}_t \notin \bigcup_{i \in [N]} \mathcal{C}_{i,t} \;\middle|\; \mathcal{F}_t\right\} = \mathbb{P}\{\tilde{p}_t \in B_{\epsilon'}(\hat{p}_t) \mid \mathcal{F}_t\}$$
$$\leq h\left(\inf_{p \in \{p: \|p - \hat{p}_t\| > \epsilon'\}} \|B_t^{1/2}(p - \hat{p}_t)\|^2\right)$$
$$\leq h\left(\lambda\|p - \hat{p}_t\|^2\right)$$
$$\leq h((\lambda\epsilon')^2) \,. \tag{119}$$

Using the definition of $\bar{i}_t$ completes the proof. $\qquad\square$

**Lemma 25.** *For any* $i \in [N]$, *the event* $\bar{\mathcal{A}}_i^c(t)$ *implies* $\|S_i\hat{p}_t - S_ip^*\| \geq \epsilon/16$.

*Proof.* Using the triangle inequality, we have
$$\|S_i\hat{p}_t - S_ip^*\| \geq \|S_i\bar{p}_t - S_ip^*\| - \|S_i\bar{p}_t - S_i\hat{p}_t\|$$
$$\geq \epsilon/8 - \|S_i\|\|\bar{p}_t - \hat{p}_t\|$$
$$\geq \epsilon/8 - \|S_i\|\frac{\epsilon}{16\max_i\|S_i\|}$$
$$\geq \epsilon/8 - \epsilon/16 = \epsilon/16 \,. \tag{120}$$
$$\square$$

*Proof of Lemma 23.* For any $n_0$, which is specified later, we have
$$\mathbb{E}\left[\sum_{t=1}^T \mathbb{1}\left[\bar{p}_t \in \mathcal{C}_i, \, \bar{\mathcal{A}}_i^c(t)\right]\right]$$
$$= \mathbb{E}\left[\sum_{t=1}^T \mathbb{1}\left[\bar{p}_t \in \mathcal{C}_i, \, \bar{\mathcal{A}}_i^c(t), \, N_i(t) < n_0\right]\right] + \mathbb{E}\left[\sum_{t=1}^T \mathbb{1}\left[\bar{p}_t \in \mathcal{C}_i, \, \bar{\mathcal{A}}_i^c(t), \, N_i(t) \geq n_0\right]\right] \tag{121}$$

The first term can be bounded by $(p_0/N)^{-1} \cdot n_0$ from Lemma 24. The rigorous proof can be obtained by the almost same argument as the following analysis of the second term using Theorem 16.

Then, we will bound the second term. Specifically, we will prove that for $n_0 = \frac{M \log T}{(\epsilon/16)^2}$,
$$\mathbb{E}\left[\sum_{t=1}^T \mathbb{1}\left[\bar{p}_t \in \mathcal{C}_i, \, \bar{\mathcal{A}}_i^c(t), \, N_i(t) \geq n_0\right]\right] = \mathrm{O}(1) \,. \tag{122}$$

First we have
$$\mathbb{E}\left[\sum_{t=1}^T \mathbb{1}\left[\bar{p}_t \in \mathcal{C}_i, \, \bar{\mathcal{A}}_i^c(t), \, N_i(t) \geq n_0\right]\right]$$
$$\leq \sum_{m=n_0}^\infty \mathbb{E}\left[\sum_{t=1}^T \mathbb{1}\left[\bar{p}_t \in \mathcal{C}_i, \, \bar{\mathcal{A}}_i^c(t), \, N_i(t) = m\right]\right] \,. \tag{123}$$

Let

$$\tau = \min\left\{t : \bar{p}_t \in \mathcal{C}_i,\ \bar{\mathcal{A}}_i^c(t),\ N_i(t) = m\right\} \wedge (T+1) \tag{124}$$

be the first time such that $\bar{p}_t \in \mathcal{C}_i$, $\bar{\mathcal{A}}_i^c(t)$ and $N_i(t) = m$ occur. Letting $\mathcal{A}_t := \left\{\bar{p}_t \in \mathcal{C}_i,\ \bar{\mathcal{A}}_i^c(t),\ N_i(t) = m\right\}$, $\mathcal{B}_t := \{i(t) = i\}$ and $P_t := p_0/N$ in Theorem 16, we have

$$\mathbb{E}\left[\sum_{t=1}^{T} \mathbb{1}\left[\bar{p}_t \in \mathcal{C}_i,\ \bar{\mathcal{A}}_i^c(t),\ N_i(t) = m\right]\right] \leq \frac{N}{p_0}\mathbb{P}\{\tau \leq T\}. \tag{125}$$

Here $\tau \leq T$ implies that

$$\begin{aligned}
\|\hat{p}_\tau - p^*\|_{B_\tau} &= (\hat{p}_\tau - p^*)^\top \left(\lambda I + \sum_{j \in [N]} N_j(\tau) S_j^\top S_j\right)(\hat{p}_\tau - p^*)\\
&\geq m(\hat{p}_\tau - p^*)^\top \left(S_i^\top S_i\right)(\hat{p}_\tau - p^*)\\
&= m\|S_i(\hat{p}_\tau - p^*)\|^2 \geq m(\epsilon/16)^2,
\end{aligned} \tag{126}$$

where the last inequality follows from Lemma 25. Therefore we have

$$\begin{aligned}
\mathbb{E}\left[\exp(\|\hat{p}_\tau - p^*\|_{B_\tau}^2/2)\right] &\geq \mathbb{E}\left[\mathbb{1}[\tau \leq T]\exp(\|\hat{p}_\tau - p^*\|_{B_\tau}^2/2)\right]\\
&\geq \exp(m(\epsilon/16)^2/2)\mathbb{P}\{\tau \leq T\}.
\end{aligned} \tag{127}$$

Note that $|B_\tau| \leq |B_T| \leq (1 + TL/M)^M$ for $L = \max_i \sqrt{\operatorname{trace}(S_i^\top S_i)} = \max_i \|S_i\|_{\mathrm{F}}$ by Lemma 10 of Abbasi-Yadkori et al. (2011), where $\|\cdot\|_{\mathrm{F}}$ is the Frobenius norm. Therefore we have

$$\begin{aligned}
\mathbb{E}\left[\exp(\|\hat{p}_\tau - p^*\|_{B_\tau}/2)\right] &\leq \mathbb{E}\left[\sqrt{|B_\tau|} \cdot \frac{\exp(\|\hat{p}_\tau - p^*\|_{B_\tau}^2/2)}{\sqrt{|B_\tau|}}\right]\\
&\leq (1 + TL/M)^{M/2}\mathbb{E}\left[\frac{\exp(\|\hat{p}_\tau - p^*\|_{B_\tau}^2/2)}{\sqrt{|B_\tau|}}\right]\\
&\leq (1 + TL/M)^{M/2}\mathbb{E}\left[\frac{\exp(\|\hat{p}_0 - p^*\|_{B_0}^2/2)}{\sqrt{|B_0|}}\right] \tag{128}\\
&= \left(1 + \frac{TL}{M\lambda}\right)^{M/2} e^{\lambda\|p^*\|^2/2}, \tag{129}
\end{aligned}$$

where (128) holds since $\frac{\exp(\|\hat{p}_t - p^*\|_{B_t}^2/2)}{\sqrt{|B_t|}}$ is a supermartingale from Lemma 12. Combining (125), (127), and (129), we obtain

$$\begin{aligned}
\sum_{m=n_0}^{\infty}\mathbb{E}\left[\sum_{t=1}^{T}\mathbb{1}\left[\bar{p}_t \in \mathcal{C}_i,\ \bar{\mathcal{A}}_i^c(t),\ N_i(t) = m\right]\right] &\leq \frac{N}{p_0}\left(\frac{1}{\lambda} + \frac{TL}{M\lambda}\right)^{M/2} e^{\lambda\|p^*\|^2/2}\sum_{m=n_0}^{\infty} e^{-m(\epsilon/16)^2/2}\\
&\leq \frac{N}{p_0}\left(\frac{1}{\lambda} + \frac{TL}{M\lambda}\right)^{M/2} e^{\lambda\|p^*\|^2/2}\frac{e^{-n_0\epsilon^2/2}}{1 - e^{-(\epsilon/16)^2/2}}.
\end{aligned} \tag{130}$$

By choosing $n_0 = \frac{M\log T}{(\epsilon/16)^2}$ we obtain the lemma. $\qquad\square$

### F.5  Analysis for Case (D)

**Lemma 26.** *For any $i \in [N]$,*

$$\mathbb{E}\left[\sum_{t=1}^{T}\mathbb{1}\left[\bar{p}_t \in \mathcal{C}_i,\ \bar{\mathcal{A}}_i(t),\ \tilde{\mathcal{A}}_i^c(t)\right]\right] \leq \frac{48}{9}\frac{M+2}{\epsilon^2}\frac{N}{p_0}. \tag{131}$$

*Remark.* To prove the regret upper bound, it is enough to prove Lemma 26 only for $i = 1$. However, for the sake of generality, we prove the lemma for any $i \in [N]$.

Before proving Lemma 26, we give two following lemmas.

**Lemma 27.** *For any $i \in [N]$, the event $\bar{\mathcal{A}}_i(t)$ implies $\mathcal{A}_i(t)$.*

*Proof.* Using the triangle inequality, we have

$$
\begin{aligned}
\|S_i p^* - S_i \hat{p}_t\| &\leq \|S_i p^* - S_i \bar{p}_t\| + \|S_i \bar{p}_t - S_i \hat{p}_t\| \\
&\leq \epsilon/8 + \|S_i\| \cdot \frac{\epsilon}{16 \max_i \|S_i\|} < \epsilon/4 \,,
\end{aligned}
\tag{132}
$$

which completes the proof. $\qquad\square$

Now, Lemma 26 can be intuitively proven because from Lemma 27, $\bar{\mathcal{A}}_i(t)$ implies $\mathcal{A}_i(t)$, and the events $\mathcal{A}_i(t)$ and $\tilde{\mathcal{A}}_i^c(t)$ does not simultaneously occur many times.

Let $t = \sigma_1, \ldots, \sigma_m$ be the time of the first $m$ times that the event $\{\bar{p}_t \in \mathcal{C}_i, \mathcal{A}_i(t), N_i(t) = n_i\}$ occurred (not $\{\bar{p}_t \in \mathcal{C}_i, \bar{\mathcal{A}}_i(t), N_i(t) = n_i\}$). In other words, we define

- $\sigma_1$ : the first time that $\bar{p}_t \in \mathcal{C}_i$, $\mathcal{A}_i(t)$ and $N_i(t) = n_i$ occurred

- $\sigma_2$ : the second time that $\bar{p}_t \in \mathcal{C}_i$, $\mathcal{A}_i(t)$ and $N_i(t) = n_i$ occurred

- ... .

Now we prove the following lemma using Lemma 9.

**Lemma 28.** *For any $0 \leq \xi < 1/2$,*

$$
\mathbb{P}\Big\{ \tilde{\mathcal{A}}_i^c(t) \,\Big|\, \mathcal{A}_i(t), \, \sigma_k = t \Big\} \leq \exp\left( -\frac{9}{16} \xi \epsilon^2 n_i \right) (1 - 2\xi)^{-M/2} \,.
\tag{133}
$$

*Proof.* Recall that $\mathcal{T}_i = \{ p \in \mathbb{R}^M : \|S_i p - S_i p^*\| > \epsilon \}$. We follow a similar argument as the analysis for Lemma 19. Since $\tilde{p}_t \sim \mathcal{N}(B_t^{-1} b_t, B_t^{-1})$, the squared Mahalanobis distance $\|B_t^{1/2}(p - \hat{p}_t)\|^2$ follows the chi-squared distribution with $M$ degree of freedom. Hence, for $h(a) = \mathbb{P}_{X \sim \chi_M^2}\{ X \geq a\}$, we have

$$
\mathbb{P}\Big\{ \tilde{\mathcal{A}}_i^c(t) \,\Big|\, \mathcal{A}_{i,n_i}, \, \sigma_k = t \Big\} \leq h\left( \inf_{p \in \mathcal{T}_i} \|B_t^{1/2}(p - \hat{p}_t)\|^2 \right) .
\tag{134}
$$

Then, Eq. (133) directly follows from Lemma 9. $\qquad\square$

*Proof of Lemma 26.* From Lemma 27, the event $\bar{\mathcal{A}}_i(t)$ implies $\mathcal{A}_i(t)$. Hence, it is enough to derive the upper bound for

$$
\mathbb{E}\left[ \sum_{t=1}^T \mathbb{1}\Big[ \bar{p}_t \in \mathcal{C}_i, \, \mathcal{A}_i(t), \, \tilde{\mathcal{A}}_i^c(t) \Big] \right]
\tag{135}
$$

instead of the bound for

$$
\mathbb{E}\left[ \sum_{t=1}^T \mathbb{1}\Big[ \bar{p}_t \in \mathcal{C}_i, \, \bar{\mathcal{A}}_i(t), \, \tilde{\mathcal{A}}_i^c(t) \Big] \right] .
\tag{136}
$$

Using Lemma 28, we can bound the term for case (D) from above as

$$\mathbb{E}\left[\sum_{t=1}^{T}\mathbb{1}\left[\bar{p}_t \in \mathcal{C}_i,\, \mathcal{A}_i(t),\, \tilde{\mathcal{A}}_i^c(t)\right]\right]$$

$$= \mathbb{E}\left[\sum_{n_i=1}^{T}\sum_{t=1}^{T}\mathbb{1}\left[\mathcal{A}_i(t),\, \tilde{\mathcal{A}}_i^c(t),\, N_i(t)=n_i\right]\right]$$

$$= \sum_{n_i=1}^{T}\sum_{t=1}^{T}\mathbb{P}\left\{\mathcal{A}_i(t),\, \tilde{\mathcal{A}}_i^c(t),\, N_i(t)=n_i\right\}$$

$$= \sum_{n_i=1}^{T}\sum_{t=1}^{T}\sum_{k=1}^{T}\mathbb{P}\left\{\mathcal{A}_i(t),\, \tilde{\mathcal{A}}_i^c(t),\, \sigma_k=t\right\} \quad \text{(the event } \{\sigma_k=t\} \text{ is exclusive for fixed } n_i)$$

$$= \sum_{n_i=1}^{T}\sum_{t=1}^{T}\sum_{k=1}^{T}\mathbb{P}\{\mathcal{A}_i(t),\, \sigma_k=t\}\mathbb{P}\left\{\tilde{\mathcal{A}}_i^c(t) \,\Big|\, \mathcal{A}_i(t),\, \sigma_k=t\right\}$$

$$\leq \sum_{n_i=1}^{T}\sum_{t=1}^{T}\sum_{k=1}^{T}\mathbb{P}\{\mathcal{A}_i(t),\, \sigma_k=t\}C\mathrm{e}^{-n_i\iota} \quad \text{(by Lemma 28)}$$

$$= \sum_{n_i=1}^{T}C\mathrm{e}^{-n_i\iota}\sum_{t=1}^{T}\sum_{k=1}^{T}\mathbb{P}\{\mathcal{A}_i(t),\, \sigma_k=t\}$$

$$\leq \sum_{n_i=1}^{T}C\mathrm{e}^{-n_i\iota}\sum_{t=1}^{T}\sum_{k=1}^{T}\mathbb{P}\{\sigma_k=t\}$$

$$\leq \sum_{n_i=1}^{T}C\mathrm{e}^{-n_i\iota}\sum_{k=1}^{T}\mathbb{P}\{\sigma_k \text{ exists}\}$$

$$\leq \sum_{n_i=1}^{T}C\mathrm{e}^{-n_i\iota}\sum_{k=1}^{T}\left(1-\frac{p_0}{N}\right)^{k-1} \quad \text{(by } \tilde{p}_{\sigma_s} \notin \mathcal{C}_i \text{ for } s=1,\ldots,k-1)$$

$$\leq 3C\frac{1}{\mathrm{e}^\iota - 1}\frac{N}{p_0}$$

$$\leq \frac{48}{9}\frac{M+2}{\epsilon^2}\frac{N}{p_0}, \tag{137}$$

where $\iota = \frac{9\xi\epsilon^2}{16}$, $C = (1-2\xi)^{-\frac{M}{2}}$, and in the last inequality we select the optimal $\xi$ and use $1+x \leq \mathrm{e}^x$. $\qquad\square$

## F.6 Analysis for Case (E)

**Lemma 29.** *For any $i \neq 1$,*

$$\mathbb{E}\left[\sum_{t=1}^{T}\mathbb{1}\left[\bar{p}_t \in \mathcal{C}_i,\, \bar{\mathcal{A}}_i(t)\right]\right] \leq \frac{2^{5M/2+7}\Gamma(M/2+1)\mathrm{e}^{\lambda^2\|p^*\|^2/2}}{\delta^2\epsilon^{M+2}\lambda^{M/2+1}}, \tag{138}$$

*where $\epsilon$ is defined in (26) and satisfies $\epsilon \leq \min_{p\in\mathcal{C}_1^c}\|p-p^*\|/3$, and*

$$\delta := \min_{\hat{p}:(L_1-L_i)^\top\hat{p}\geq 0,\, \|S_i(\hat{p}-p^*)\|\leq\epsilon/8}\|S_1(\hat{p}-p^*)\|. \tag{139}$$

We prove Lemma 29 using Lemma 12 and Theorem 16.

*Remark.* The upper bound in (138) goes to infinite when we set $\lambda=0$, that is, a flat prior is used. However, this not the essential effect of the prior but just comes from the minimum eigenvalue of $B_1$. In fact, we can see from the proof that a similar bound can be obtained for $\lambda=0$ if we run some deterministic initialization until $B_t$ becomes positive definite.

*Proof.* We evaluate each term in the summation using Theorem 16 with

$$\mathcal{A}_t = \{\bar{p}_t \in \mathcal{C}_i, \ \|S_i(\bar{p}_t - p^*)\| \leq \epsilon/8, \ N_1(t) = n\},$$
$$\mathcal{B}_t = \{\tilde{p}_t \in \mathcal{C}_1\}. \tag{140}$$

for $n \in [T]$. Recall that

$$\bar{g}_t(p) = \frac{1}{\sqrt{(2\pi)^M |B_t^{-1}|}} \exp\left(-\frac{1}{2} \|p - \hat{p}_t\|_{B_t}^2\right) \tag{141}$$

is the probability density function of $\hat{p}_t$ given $\mathcal{F}_t = \{B_t, b_t\}$. Using $\tau$ defined in (80), it holds for any $\tau \in [T]$ that

$$\mathbb{P}\{\mathcal{B}_\tau | \mathcal{F}_\tau\} = \mathbb{P}\{\tilde{p}_\tau \in \mathcal{C}_1 \mid \mathcal{F}_\tau\}$$

$$= \int_{p \in \mathcal{C}_1} \bar{g}_\tau(p) \mathrm{d}p$$

$$\geq \int_{p:\|p-p^*\| \leq 3\epsilon} \bar{g}_\tau(p) \mathrm{d}p$$

$$\geq \sup_{p:\|p-p^*\| \leq 2\epsilon} \int_{p':\|p'-p\| \leq \epsilon} \bar{g}_\tau(p') \mathrm{d}p' \tag{142}$$

$$\geq \sup_{p:\|p-p^*\| \leq 2\epsilon} \inf_{p':\|p'-p\| \leq \epsilon} \bar{g}_\tau(p') \mathrm{Vol}(\{p'' : \|p'' - p\| \leq \epsilon\})$$

$$= \frac{(\sqrt{\pi}\epsilon)^M}{\Gamma(M/2+1)} \sup_{p:\|p-p^*\| \leq 2\epsilon} \inf_{p':\|p'-p\| \leq \epsilon} \bar{g}_\tau(p')$$

$$= \frac{(\epsilon/\sqrt{2})^M \sqrt{|B_\tau|}}{\Gamma(M/2+1)} \exp\left\{-\frac{1}{2}\left(\inf_{p:\|p-p^*\| \leq 2\epsilon} \sup_{p':\|p'-p\| \leq \epsilon} \|p' - \hat{p}_\tau\|_{B_\tau}^2\right)\right\}$$

$$\geq \frac{(\epsilon/\sqrt{2})^M \sqrt{|B_\tau|}}{\Gamma(M/2+1)} \exp\left\{-\frac{\|\hat{p}_\tau - p^*\|_{B_\tau}^2 - \epsilon\delta\sqrt{\lambda n}}{2}\right\}, \tag{143}$$

where (142) follows since $\{p : \|p-p^*\| \leq 3\epsilon\} \supset \{p' : \|p'-p_0\| \leq \epsilon\}$ for any $p_0$ such that $\|p_0 - p^*\| \leq 2\epsilon$, and the last inequality follows from Theorem 15. To apply Theorem 15, we used Lemma 27.

Now we define a stochastic process corresponds to (143) as

$$P_t = \frac{(\epsilon/\sqrt{2})^M \sqrt{|B_t|}}{\Gamma(M/2+1)} \exp\left\{-\frac{\|\hat{p}_t - p^*\|_{B_t}^2 - \epsilon\delta\sqrt{\lambda n}}{2}\right\}. \tag{144}$$

Then, by Lemma 12,

$$\mathbb{E}[P_{t+1}^{-1}|\mathcal{F}_t] \leq \frac{\Gamma(M/2+1)}{(\epsilon/\sqrt{2})^M} \mathrm{e}^{-\epsilon\delta\sqrt{\lambda n}/2} \mathbb{E}\left[\frac{1}{\sqrt{|B_{t+1}|}} \mathbb{E}\left[\exp\left(\frac{\|\hat{p}_t - p^*\|_{B_{t+1}}^2}{2}\right) \middle| \mathcal{F}_t, S_{i(t)}\right] \middle| \mathcal{F}_t\right]$$

$$\leq \frac{\Gamma(M/2+1)}{(\epsilon/\sqrt{2})^M} \mathrm{e}^{-\epsilon\delta\sqrt{\lambda n}/2} \mathbb{E}\left[\frac{1}{\sqrt{|B_t|}} \exp\left(\frac{\|\hat{p}_t - p^*\|_{B_t}^2}{2}\right) \middle| \mathcal{F}_t\right]$$

$$= P_t^{-1}, \tag{145}$$

which means that $P_t^{-1}$ is a supermartingale. Therefore we can apply Theorem 16 and obtain

$$\mathbb{E}\left[\sum_{t=1}^{T} \mathbb{1}[\hat{p}_t \in \mathcal{C}_i, \ \|S_i(\bar{p}_t - p^*)\| \leq \epsilon/8, \ N_1(t) = n]\right] \leq \mathbb{E}\left[\mathbb{1}[\tau \leq T]P_\tau^{-1}\right]$$

$$\leq \mathbb{E}\left[P_\tau^{-1}\right]$$

$$\leq \mathbb{E}\left[P_1^{-1}\right]$$

$$= \frac{\Gamma(M/2+1)\mathrm{e}^{\lambda^2\|p^*\|^2/2}}{(\epsilon\sqrt{\lambda/2})^M} \mathrm{e}^{-\epsilon\delta\sqrt{\lambda n}/2}. \tag{146}$$

Finally we have

$$
\mathbb{E}\left[\sum_{t=1}^{T}\mathbb{1}[\bar{p}_t \in \mathcal{C}_i,\ \|S_i(\bar{p}_t - p^*)\| \le \epsilon/8]\right]
$$

$$
= \sum_{n=1}^{T}\mathbb{E}\left[\sum_{t=1}^{T}\mathbb{1}[\bar{p}_t \in \mathcal{C}_i,\ \|S_i(\bar{p}_t - p^*)\| \le \epsilon/8,\ N_1(t)=n]\right]
$$

$$
\le \frac{\Gamma(M/2+1)\mathrm{e}^{\lambda^2\|p^*\|^2/2}}{(\epsilon\sqrt{\lambda/2})^M}\sum_{n=1}^{\infty}\mathrm{e}^{-\epsilon\delta\sqrt{\lambda n}/2}
$$

$$
\le \frac{\Gamma(M/2+1)\mathrm{e}^{\lambda^2\|p^*\|^2/2}}{(\epsilon\sqrt{\lambda/2})^M}\int_{0}^{\infty}\mathrm{e}^{-\epsilon\delta\sqrt{\lambda x}/2}\mathrm{d}x
$$

$$
= \frac{\Gamma(M/2+1)\mathrm{e}^{\lambda^2\|p^*\|^2/2}}{(\epsilon\sqrt{\lambda/2})^M}\frac{2}{(\epsilon\delta\sqrt{\lambda}/2)^2}\Gamma(2)
$$

$$
= \frac{2^{M/2+3}\Gamma(M/2+1)\mathrm{e}^{\lambda^2\|p^*\|^2/2}}{\delta^2\epsilon^{M+2}\lambda^{M/2+1}}, \tag{147}
$$

which completes the proof. $\qquad\square$

## G  Property of Dynamic Pricing Games

In this appendix, we will see a property of dp-easy games.

**Proposition 30.** *Consider any dp-easy games with $c > -1$. Then, any two actions in the game are neighbors.*

*Remark.* In section 5, we considered dp-easy games with $c > 0$, but this can be relaxed to $c > -1$ to prove Proposition 30.

*Proof.* Take any two different actions $j, k \in [N]$ such that $j < k$. From the definition of the loss matrix in dp-easy games, we have $e_j \in \mathcal{C}_j$ and $e_k \in \mathcal{C}_k$.

First, we will find $\alpha \in [0, 1]$ such that

$$
\alpha e_j + (1-\alpha)e_k \in \mathcal{C}_j \cap \mathcal{C}_k. \tag{148}
$$

From the definition of the loss matrix, the $i$-th element of $L(\alpha e_j + (1-\alpha)e_k) \in \mathcal{P}_M$ is

$$
\begin{cases}
-i & (1 \le i \le j) \\
\alpha c + (1-\alpha)\cdot(-i) & (j+1 \le i \le k) \\
c & (k < i \le N)
\end{cases} \tag{149}
$$

It is easy to see that the indices which give the minimum value in (149) is $j$ or $k$. Thus, to achieve the condition (148), the following should be satisfied,

$$
-j = \alpha c + (1-\alpha)\cdot(-k), \tag{150}
$$

which is equivalent to

$$
\alpha = \frac{k-j}{c+k}(=: \alpha^*). \tag{151}
$$

Note that we have $0 \le \alpha \le 1$ for any $c > -1$.

Next, we introduce the following definitions.

$$
p^{(j,k)} := \alpha^* e_j + (1-\alpha^*)e_k \in \mathcal{C}_j \cap \mathcal{C}_k, \tag{152}
$$

$$
\mathrm{Ball}_\epsilon^{(j,k)} := \left\{p \in \mathcal{P}_M : \|p - p^{(j,k)}\| \le \epsilon\right\}, \tag{153}
$$

$$
L^{(x)} := L(p^{(j,k)} + x) \in \mathbb{R}^N. \tag{154}
$$

To prove the proposition, it is enough to prove the following: there exists $\epsilon > 0$, $\mathrm{Ball}_\epsilon^{(j,k)} \subset \mathcal{C}_j \cup \mathcal{C}_k$.

To prove this, it is enough to prove that, there exists $\epsilon > 0$,

$$\min_{x \in \mathbb{R}^M : \|x\| \leq \epsilon} \min_{i \in [N]\backslash\{j,k\}} \left( (L^{(x)})_i - (L^{(x)})_j \right) \vee \left( (L^{(x)})_i - (L^{(x)})_k \right) > 0. \tag{155}$$

We will prove (155) in the following. Take any $i \in [N]\backslash\{j,k\}$ and

$$\epsilon := \min_{i:1 \leq i < j} \frac{1}{2} \frac{j-i}{\|L_j - L_i\|} \wedge \min_{i:j < i < k} \frac{1}{2} \frac{(1-\alpha^*)(k-i)}{\|L_i - L_k\|} \wedge \min_{i:k < i \leq N} \frac{1}{2} \frac{c+j}{\|L_j - L_i\|}. \tag{156}$$

Note that the $\epsilon$ used here is different from the one used in the proof of the regret upper bounds.

**Case (A):** When $1 \leq i < j$, using Cauchy–Schwarz inequality, we have

$$
\begin{aligned}
\left( (L^{(x)})_i - (L^{(x)})_j \right) \vee \left( (L^{(x)})_i - (L^{(x)})_k \right) &\geq (L^{(x)})_i - (L^{(x)})_j \\
&= (-i + L_i^\top x) - (-j + L_j^\top x) \\
&= (j-i) - (L_j - L_i)^\top x \\
&\geq (j-i) - \|L_j - L_k\|\|x\| \\
&\geq (j-i) - \epsilon\|L_j - L_i\| \\
&\geq \frac{1}{2}(j-i) \\
&> 0.
\end{aligned} \tag{157}
$$

The arguments for cases (B) and (C) follow in the similar manner as case (A).

**Case (B):** When $j < i < k$, we have

$$
\begin{aligned}
\left( (L^{(x)})_i - (L^{(x)})_j \right) \vee \left( (L^{(x)})_i - (L^{(x)})_k \right) &\geq (L^{(x)})_i - (L^{(x)})_k \\
&= \{\alpha^* c + (1-\alpha^*)\cdot(-i) + L_i^\top x\} - \{\alpha^* c + (1-\alpha^*)\cdot(-k) + L_k^\top x\} \\
&= (1-\alpha^*)(k-i) - (L_i - L_k)^\top x \\
&\geq (1-\alpha^*)(k-i) - \epsilon(L_i - L_k)^\top \\
&\geq \frac{1}{2}(1-\alpha^*)(k-i) \\
&> 0.
\end{aligned} \tag{158}
$$

**Case (C):** When $k < i \leq N$, we have

$$
\begin{aligned}
\left( (L^{(x)})_i - (L^{(x)})_j \right) \vee \left( (L^{(x)})_i - (L^{(x)})_k \right) &\geq (L^{(x)})_i - (L^{(x)})_j \\
&= (c + L_i^\top x) - (-j + L_j^\top x) \\
&\geq c + j - \|L_j - L_i\|\|x\| \\
&\geq c + j - \epsilon\|L_j - L_i\| \\
&\geq \frac{1}{2}(c+j) \\
&> 0.
\end{aligned} \tag{159}
$$

Summing up the argument for cases (A) to (C), the proof is completed. $\qquad\square$

## H   Details and Additional Results of Experiments

Here we give the specific values of the opponent's strategy used in Section 5 and show the extended experimental results for performance comparison. Table 2 summarizes the values of opponent's strategy used in this appendix and Section 5. Figure 3 shows the empirical comparison of the proposed algorithms against the benchmark methods, and Figure 4 shows the number of the rejected times. We can see the same tendency as Section 5, that is, TSPM performs the best and the number of rejections does not increase with the time step $t$.

**Table 2:** The values of the opponent's strategy.

| # of outcomes $M$ | opponent's strategy $p^*$ |
|---|---|
| 2 | $[0.7, 0.3]$ |
| 3 | $[0.5, 0.3, 0.2]$ |
| 4 | $[0.3, 0.3, 0.3, 0.1]$ |
| 5 | $[0.2, 0.3, 0.3, 0.1, 0.1]$ |
| 6 | $[0.2, 0.2, 0.3, 0.1, 0.1, 0.1]$ |
| 7 | $[0.2, 0.2, 0.3, 0.1, 0.1, 0.05, 0.05]$ |

(a) dp-easy, $N = M = 2$     (b) dp-easy, $N = M = 3$     (c) dp-easy, $N = M = 4$

(d) dp-easy, $N = M = 5$     (e) dp-easy, $N = M = 6$     (f) dp-easy, $N = M = 7$

(g) dp-hard, $N = M = 2$     (h) dp-hard, $N = M = 3$     (i) dp-hard, $N = M = 4$

(j) dp-hard, $N = M = 5$     (k) dp-hard, $N = M = 6$     (l) dp-hard, $N = M = 7$

**Figure 3:** Regret-round plots of the algorithms. The solid lines indicate the average over 100 independent trials. The thin fillings are the standard error.

(a) dp-easy, $N = M = 2$      (b) dp-easy, $N = M = 3$      (c) dp-easy, $N = M = 4$

(d) dp-easy, $N = M = 5$      (e) dp-easy, $N = M = 6$      (f) dp-easy, $N = M = 7$

(g) dp-hard, $N = M = 2$      (h) dp-hard, $N = M = 3$      (i) dp-hard, $N = M = 4$

(j) dp-hard, $N = M = 5$      (k) dp-hard, $N = M = 6$      (l) dp-hard, $N = M = 7$

**Figure 4:** The number of rejected times by the accept-reject sampling. The solid lines indicate the average over 100 independent trials after taking moving average with window size 100.