[Reviews · NeurIPS 2020]

Review 1

Summary and Contributions: This paper studies the stochastic partial monitoring (PM) problem using Thompson sampling (TS). The author designs a new TS-based algorithm that can sample from the exact posterior distribution. On the theoretical side, the paper provides the first logarithmic regret bound for a TS-based algorithm for linear bandit problems and strongly locally observable PM games. Numerically, simulations show that the proposed algorithm outperforms the previous methods.

Strengths: The paper provides the first theoretical guarantee that achieves logarithmic regret bound using TS-based sampling. Furthermore, the author performs experiments and show the advantage of the new algorithm over the existing methods.

Weaknesses: This main weakness of this paper is that it only focuses on the finite stochastic PM games while other general settings such as sub-Gaussian is of great importance.

Correctness: The technical part of this paper looks correct to me. The experimental part also looks solid. However, I didn't get a chance to verify the details in the supplementary materials.

Clarity: The paper is nicely written with a comprehensive overview of the problem and clearly presented technical ingredients.

Relation to Prior Work: The paper clearly addresses related works on PM and other methods of sampling from the posterior distributions. There are only a few previous works on analyzing TS and PM due to the intrinsic difficulty of the problem.

Reproducibility: Yes

Additional Feedback: After the response phase, considering the additional feedback received, I remain with my initial assessment of the paper.


Review 2

Summary and Contributions: This paper considers the application of Thompson Sampling (TS) to stochastic finite partial monitoring (PM). The authors propose a TS scheme using accept-reject sampling, with a parameter which can be tuned to trade-off the accuracy of the sampler for improved computational efficiency. The authors then introduce a "linearized" variant of the finite PM problem where the observed signals have additional (unit variance) Gaussian noise. One feature of this variant is that the accept-reject routine is not required as the posterior matches the proposal of the accept-reject scheme. The authors derive an instance-dependent O(log(T)) bound on the regret of their TS approach for this linearized problem (not the standard finite partial monitoring framework). In experiments, the proposed TS compares favourably to existing stochastic PM algorithms of Piccoloboni and Schindlehauer (2001) and Vanchinathan et al. (2014).

Strengths: The paper gives an interesting and new result. The understanding of TS for PM is, as the authors point out, presently limited and this analysis improves our understanding of the extent to which it is effective. This insight is relevant to the NeurIPS community. While I have not examined the entire supplement in rigorous detail, it seems that the proof uses interesting ideas, nicely exposited on lines 251-262. Further the experiments display that the method considered substantially improves over the competitors that were investigated.

Weaknesses: I think the makings of a very good paper are here, but there are several ways in which it could be improved. Firstly, there is the issue of the mismatch between the setting of the theoretical results and the experimental section/broader focus of the paper. The algorithm is initially posed in a different setting (called "discrete setting") to the theoretical results, and then the experimental section also, as I understand, considers the discrete setting. Why not make the entire focus of the paper the setting where theoretical results can be obtained? Further, around this point, it could be clearer why results cannot be obtained in the discrete setting, and why the noise in the linearized setting needs to have unit variance as described on line 200. If this is the case the scope of the theoretical contribution seems to be somewhat limited. Second, I believe that the Mario Sampling algorithm of Lattimore and Szepesvari (2019) "An Information-Theoretic Approach to Minimax Regret in Partial Monitoring" would be applicable in this setting. The experimental section therefore suffers from incompleteness by not acknowledging this approach. A more complete picture of the performance of TS relative to other approaches would be provided by also comparing to this scheme. Finally, I feel that while the results are themselves undoubtedly interesting to the NeurIPS community, the size of the contribution is perhaps a bit smaller than the average good NeurIPS paper. As said previously the setting in which the results hold feels somewhat limited. If results for the discrete setting could be established, or further insight as to either why this is challenging were reported, for instance, this would represent a more substantial contribution.

Correctness: I was not able to find issues with the correctness of the work.

Clarity: Yes, the paper is mostly well written. The authors display a clear knowledge of theoretical analysis of partial monitoring and cover the key concepts. There are areas where the writing could be more purposeful. For instance in lines 43-45, some papers containing algorithms are listed but there is no real insight as to what these algorithms entail. Similarly, in lines 130-139 a number of sampling procedures are mentioned, but without any real discussion of their relative pros or cons.

Relation to Prior Work: As I mention in the weaknesses section, I think there is a missing comparison to the PM algorithm of Lattimore and Szepesvari. Less critically, I think the discussion of existing results around TS is missing a line of work following from Russo and Van Roy's information-theoretic analysis of the Bayesian regret of TS. A fuller picture of existing results on the performance of TS could be given by reviewing this line of work.

Reproducibility: Yes

Additional Feedback: UPDATE: Thank you for your response and effectively addressing my concerns. In light of the rebuttal, other reviews and some reflection I have increased my score. I think that there is a sufficient amount of interesting content to merit publication. Some minor comments: In the comments on the upper bound it would be useful to have some sense of the magnitude of the z_{j,k} terms. In line 216 it is unclear whether the algorithm is different from the family of TS used in practical settings because of the aforementioned dependence on the time horizon, or some other reason. Lines 244-250 are not the clearest to the reader. Eventually it becomes clear that this is a discussion of the mechanisms used in the proof, but this could be established more clearly.


Review 3

Summary and Contributions: The subject of the paper is finite stochastic partial monitoring (PM): at each stage an agent chooses an action in a finite set, an opponent does likewise in another finite set (according to an unknown probability distribution) and the agent gets to observe a signal depending on both actions. the agent also incurs an unobserved loss depending on these same actions. The setting of discrete PM is introduced (where the number of signals is finite) and a thompson-sampling based algorithm is presented. That algorithm uses an accept-reject procedure to sample from the posterior by using samples from a Gaussian auxiliary distribution. The algorithm is analysed in a different setting, not discrete PM but linear PM, in which the signals are not discrete but have a linear structure. A O(log T) expected regret is shown. As a particular case, this is the first log(T) regret bound for Thompson sampling for linear bandits.

Strengths: The algorithm is well described, uses an interesting method to sample exactly from the posterior, and the authors obtain the first regret bounds for a Thompson sampling approach in that setting. The empirical performance of the algorithm on the discrete PM problem is much better than existing approaches. The first logarithmic bound for Thompson sampling for linear bandits is a significant contribution to the bandit literature.

Weaknesses: The bound is only valid for the linear PM problem with Gaussian noise, and not for sub-Gaussian noise as is often the case with bandit algorithms (which would also include the discrete PM case). From the experimental evaluation, it looks like the rejection sampling procedures suffers from many rejects when the number of actions gets larger, such that the approach becomes less practical. This is not discussed in the paper.

Correctness: I did not read the entirety of the analysis in the appendix but only checked a few points. What I checked was correct.

Clarity: The introduction of the setting of discrete PM and the description of the algorithm are very clear. The experiments are well presented as well. The setting changes suddenly at the bottom of page 5 from discrete partial monitoring to Gaussian linear. The transition is abrupt and not well hinted at before. Indeed there is even the misleading statement of line 145: "we also give theoretical analysis for the proposed algorithm". This is not true in the discrete setting under discussion at that point. A detail: in definition 3, k is a symbol, while in definition 4 it is an action. Clarity would be improved by associating more strongly notations and concepts.

Relation to Prior Work: Prior work is clearly discussed.

Reproducibility: Yes

Additional Feedback: Line 287: "Note that our algorithm can also be applied to a hard game, though there is no theoretical guarantee". Since the problems used in the experimental section are of the discrete type, there is also no guarantee for the easy case.


Review 4

Summary and Contributions: The paper studies stochastic bandits with a finite set of actions. It makes two novelty claims: 1. An argument that Thompson sampling achieves gap-dependent bound of sum(Delta_i / Delta_min^2 * log(T)). 2. An approximation algorithm of Thompson sampling posteriors with speed-performance trade-offs. === After reading the other reviews and author feedback, I feel more associated with the motivations in this work. I want to see two points of improvements before camera ready: 1. The introduction is lacking concrete examples. What is partial monitoring? Why is it important? What applications would this lead to? Research contributions should be measured by the amount of new opportunities it may generate. To give readers room for imagination, one strategy would be to provide numerical examples right after each motivation claim. 2. The sampling strategy seems to have some limitations due to curse of dimensionality. I would like to see a discussion of the limitations and why that does not matter in actual application of this work. Good writing should make their work motivating, accessible, and enjoyable. Even in pure math, people still find direct applications, e.g., in cryptography, to make their impacts obvious. In the latter, people may make a bigger effort to appreciate the work because they feel associated.

Strengths: Relevance to the NeurIPS community: Bandits algorithm and analysis is of interest to the community. Clarity: The paper has a clear organization.

Weaknesses: Novelty: The claim #1 is expected from very early results. The claim #2 is unclear because the number of rejection times seems high, which indicates that the approximation algorithm may be inefficient. Empirical: The paper includes experiments, but they are limited in scale. The lack of contextual feature modeling also makes the work difficult to apply in practice.

Correctness: I did not read the appendix but the result agrees with my expectations based on related work.

Clarity: Yes

Relation to Prior Work: No. I would expect to see three points of clarifications: 1. The main theoretical novelty came from a claim that a high-probably regret bound cannot yield a regret bound in expectation (line 223). This is surprising. 2. The complaint against "a time-consuming optimization problem" needs to be elaborated. Particularly, how does it relate to the objective function (1)? 3. The discussion about Bartok et al., 2011 is unspecific. The author aims to classify minimax regrets to trivial, easy, hard, and hopeless, but did not give any intuitions.

Reproducibility: Yes

Additional Feedback: * I am not seeing the point to mention PM games. All bandits work can be framed as minimax games. How is this work different? This confused me a little in the introduction. * Definition 1 through Definition 5 could be mostly deleted without impairing understanding. What is a Pareto front doing here? * Let me know if I understood the contributions correctly. If so, Section 3.1 needs to be clarified. For example, the rejection function is not presented. * Section 3.2 seems intuitively useful, but it is completely out of context. I am not clear if it makes a difference with or without rank reductions. * Experiments setup. Needs to remind reader what is N and M. (I guess number of arms?) How can they be different? * There is a general lack of discussions of the results - how different conditions impact the cumulative regrets. The image backgrounds also should be set as white or transparent. * Could you add the theoretical regret curve in this experiment for comparisons? If it incurs large constants, then you could draw it in a twinx plot. I am primarily hoping to see the logarithmic rates.

[Author Response · NeurIPS 2020]

We thank all the reviewers for their constructive comments. We address the main points of the reviews in the following.
We will also address other comments in the revised version. We abbreviate Thompson sampling and partial monitoring
to TS and PM, respectively.

**To Reviewers #1, #2, and #3 (on Generalization to Sub-Gaussian Noise)**

The restriction to the Gaussian noise comes from the essential difficulty of the problem-dependent analysis of TS,
where lower bounds for some probabilities are needed whereas the sub-Gaussian assumption is suited for obtaining
upper bounds. In fact, to the best of our knowledge, the problem-dependent regret analysis for TS on the sub-Gaussian
case has never been investigated even for the multi-armed bandit setting, which is quite simple compared to that of
PM. In the literature, the noise distribution is restricted to distributions with explicitly given forms, e.g., Bernoulli,
Gaussian, or more generally a one-dimensional canonical exponential family (Kaufmann et al., 2012; Agrawal and
Goyal, 2013a, Korda et al., 2013). Their analysis relies on the specific characteristic of the distribution to bound the
problem-dependent regret. We will add this discussion in the revised version.

(Korda et al., 2013) Thompson Sampling for 1-Dimensional Exponential Family Bandits, In NeurIPS2013.

**To Reviewer #2**

> "Why not make the entire focus of the paper the setting where theoretical results can be obtained?"
The linear PM has been mainly considered from the theoretical viewpoint and experiments have not been conducted.
Therefore, we conducted experiments on the discrete setting for fair comparison with existing work.

> "In line 216 it is unclear whether the algorithm is different from the family of TS used in practical settings"
Their algorithm considers the posterior distribution for *regret* (not pseudo-regret), and an action is chosen according to
the posterior probability that each arm minimizes the *cumulative* regret (thus the time horizon also needs to be known),
whereas the typical TS considers the pseudo-regret at each round. We will make it more clear in the revised version.

> "The experimental section ... incompleteness by not acknowledging this approach (= Mario sampling)."
Thank you for pointing out the lack of reference to the important work. Mario sampling is the algorithm for locally
observable games and not applicable to hard games, like dp-hard. On the other hand for locally observable games,
Mario sampling coincides with TS (except for the above difference between pseudo-regret and regret with known
time horizon) when any pair of actions is a neighbor. We confirmed that some dp-easy games satisfy this property,
and conjecture that it generally holds for dp-easy. Therefore, the performance is essentially the same between TSPM
($R = 1$) and Mario sampling, though general analysis on the difference is an important future direction.

> "In the comments on the upper bound it would be useful to have some sense of the magnitude of the $z_{j,k}$ terms."
Intuitively, the norm of $z_{j,k}$ indicates the difficulty of the problem. Whereas we can estimate $(S_j p, S_k p)$ with noise
through taking actions $j$ and $k$, the actual interest is the gap of the losses $p^\top (L_j - L_k) = (S_j p, S_k p)^\top z_{j,k}$. Thus,
if $\|z_{j,k}\|$ is large, the gap estimation becomes difficult since the noise is enhanced through $z_{j,k}$. We will add this
discussion in the revised version.

**To Reviewer #3 and Reviewer #4 (on the Number of Rejected Times in Accept-Reject Sampling)**

In the accept-reject sampling, it is desirable that the frequency of rejection (a) does not increase as the time-step $t$
and (b) does not increase so much with the number of outcomes. From the experimental results, we can see that the
property (a) is indeed satisfied. For the property (b), it is true that the frequency of rejection becomes large when exact
sampling ($R = 1$) is conducted, as pointed by R3. Still, we can substantially improve this frequency by setting $R$ to be a
small value or zero, which still keeps regret tremendously better than that of BPM with almost the same time-efficiency
as BPM-TS. This result exhibits a clear speed-performance trade-off. We will make it more clear in the revised version.

**To Reviewer #4**

> "... The paper includes experiments, but they are limited in scale. The lack of contextual feature modeling ... "
The scale of the experiment is determined based on standard literature (Bartók et al., 2012; Komiyama et al., 2015).
The analysis of the contextual and the non-contextual settings is essentially different, because achievable regrets and
appropriate algorithms can be different between them.

> "Experiments setup. Needs to remind reader what is N and M. (I guess number of arms?) How can they be different?"
Thank you for the suggestion. In the revised version, we will explicitly describe that "price" and "evaluation value"
correspond to the action and the outcome, respectively.

> "There is a general lack of discussions of the results - how different conditions impact the cumulative regrets."
The discussion on the performance comparison of methods and the rejection sampling is given in Line 297–306. In
addition to that, we can say that the proposed methods outperform BPM-TS more significantly for a larger number of
outcomes. This can be seen from the discussion in Appendix D, and we will make it more clear in the revised version.

[Meta-Review · NeurIPS 2020]

This paper generated a lot of discussion amongst the reviewers. There is largely agreement that this work contains many interesting results, and I agree. I also agree with the reviewers that there is some discontinuity between the experiments and the theory and encourage the authors to use their extra page to carefully explain the choices and differences between the set-ups. Ultimately I do think it is nice to see experimental evidence in the discrete setting, which possibly hints towards theoretical results in that setting as well. One question I had while reading the paper: Do you think there is hope to prove a sqrt(T) minimax bound for TS for strongly observable games? The logarithmic regret bound that you prove would normally lead to a T^{2/3} bound, since it depends on 1/Delta^2. Some discussion on this might be interesting. I also found a few minor typos. L21: "unaware" -> "unobserved" L24: "the whole rounds" -> "all rounds"? L99: The definition of Pareto optimal may be not quite correct when there are duplicate actions (actions with the same loss). L102: In the neighbour definition the union should be an intersection. L231: The minimax version of this bound will be O(T^{2/3}) while O(T^{1/2}) is known to be possible in this setting. Do you have any sense about whether or not this is appearing only in the analysis or also the algorithm? * How does this algorithm compare to IDS by Kirschner and Lattimore in the linear setting? L287: Do you believe a theoretical guarantee is possible? I would guess not. E.g., what happens in a game where there is a costly, but informative action and all other actions give no information. Will not Thompson sampling quickly learn not to play the informative action and then suffer linear regret? L286: Do you explicitly say that dp-easy is (strongly) locally observable and dp-hard is only globally observable? L209: Since the unknown parameter is now in R^M, how do the definitions of local observability and the cell decomposition need to change?